# Metabolic priming by multiple enzyme systems supports glycolysis, HIF1α stabilisation, and human cancer cell survival in early hypoxia

Fiona Grimm[1], Agustín Asuaje [ID][1], Aakriti Jain [ID][1], Mariana Silva dos Santos [ID][2], Jens Kleinjung[3],
Patrícia M Nunes[1], Stefanie Gehrig[1], Louise Fets [ID][1], Salihanur Darici[1], James I MacRae [ID][2] &
Dimitrios Anastasiou [ID][1]✉

## Abstract

**Adaptation to chronic hypoxia occurs through changes in protein expression, which are controlled by hypoxia-inducible factor 1α (HIF1α) and are necessary for cancer cell survival. However, the mechanisms that enable cancer cells to adapt in early hypoxia, before the HIF1α-mediated transcription programme is fully established, remain poorly understood. Here we show in human breast cancer cells, that within 3 h of hypoxia exposure, glycolytic flux increases in a HIF1α-independent manner but is limited by NAD⁺ availability. Glycolytic ATP maintenance and cell survival in early hypoxia rely on reserve lactate dehydrogenase A capacity as well as the activity of glutamate-oxoglutarate transaminase 1 (GOT1), an enzyme that fuels malate dehydrogenase 1 (MDH1)-derived NAD⁺. In addition, GOT1 maintains low α-ketoglutarate levels, thereby limiting prolyl hydroxylase activity to promote HIF1α stabilisation in early hypoxia and enable robust HIF1α target gene expression in later hypoxia. Our findings reveal that, in normoxia, multiple enzyme systems maintain cells in a primed state ready to support increased glycolysis and HIF1α stabilisation upon oxygen limitation, until other adaptive processes that require more time are fully established.**

**Keywords** Hypoxia; Glycolysis; HIF1α; Metabolism; α-Ketoglutarate
**Subject Categories** Cancer; Metabolism

## Introduction

Glucose metabolism is important for many physiological and pathological processes. In cancer, glycolysis is often increased though multiple mechanisms that include upregulated expression of glucose transporters and glycolytic genes, differential expression of metabolic enzyme isoforms, and aberrant oncogenic signalling (Gambhir et al, 2001; Gatenby and Gillies, 2004; Hsu and Sabatini, 2008). These mechanisms promote both glucose transport into cells as well as increased glycolytic enzyme activity, which, collectively, enhance glucose metabolism (Porporato et al, 2011). Furthermore, a high oxidised-to-reduced nicotinamide adenine dinucleotide (NAD⁺/NADH) ratio is also required to sustain the NAD⁺-dependent activity of the glycolytic enzyme glyceraldehyde 3-phosphate dehydrogenase (GAPDH), which becomes rate-limiting when glycolysis is high (Hosios and Vander Heiden, 2018; Liberti et al, 2017).

The function of increased glycolysis in tumours remains under intense investigation. Although glucose metabolism can provide precursors for biosynthetic pathways, a relatively low proportion of glucose carbons enters biomass production (Hosios et al, 2016). However, there is significant evidence that a major role of glycolysis is to maintain energy balance by producing ATP under conditions that perturb cellular bioenergetics (DeBerardinis and Chandel, 2016; Hao et al, 2010; Kroemer and Pouyssegur, 2008). Accordingly, increased glucose uptake correlates well with hypoxic tumour regions (Cher et al, 2006; Gatenby and Gillies, 2007; van Baardwijk et al, 2007), where ATP production from mitochondria is attenuated due to limiting oxygen (Bunn and Poyton, 1996; Gerweck et al, 1993).

During chronic hypoxia, upregulation of glycolysis is achieved through a co-ordinate increase in the activities of glycolytic enzymes (Robin et al, 1984) linked to an increased expression of the corresponding genes that is orchestrated by the prolyl hydroxylase (PHD)-hypoxia-inducible factor 1α (HIF1α) signalling axis (Eales et al, 2016; Seagroves et al, 2001; Wheaton and Chandel, 2011). PHDs use iron, α-ketoglutarate (αKG), ascorbate and $O_2$ to hydroxylate proline residues on HIF1α, which are then recognised by the E3 ligase von Hippel Lindau (VHL), leading to its ubiquitination and subsequent degradation by the proteasome (Berra et al, 2003; Bruick and McKnight, 2001; Chowdhury et al, 2009; Epstein et al, 2001; Ivan and Kaelin, 2017; Maxwell et al, 1999). The $K_m$ of PHDs for $O_2$ lies within the physiologically

[1]Cancer Metabolism Laboratory, The Francis Crick Institute, 1 Midland Road, NW1 1AT London, UK. [2]Metabolomics Science Technology Platform, The Francis Crick Institute, 1 Midland Road, NW1 1AT London, UK. [3]Computational Biology Science Technology Platform, The Francis Crick Institute, 1 Midland Road, NW1 1AT London, UK.
✉E-mail: dimitrios.anastasiou@crick.ac.uk

relevant range of $O_2$ concentrations in tissues, therefore these enzymes are thought to function as oxygen sensors (Chan et al, 2016; Ehrismann et al, 2007; Ivan et al, 2001; Koivunen et al, 2006; Wang et al, 1995). Binding of $O_2$ to the catalytic pocket of PHDs requires prior binding of αKG (Chowdhury et al, 2009), which also prevents re-association of hydroxylated HIF1α to PHDs, enabling more efficient HIF1α degradation (Abboud et al, 2018). Some oncometabolites can outcompete αKG binding to PHDs, leading to HIF stabilisation (Hewitson et al, 2007; Intlekofer et al, 2017; Koivunen et al, 2007; Selak et al, 2005), and this can be alleviated by exogenous αKG (Isaacs et al, 2005; MacKenzie et al, 2007; Tennant et al, 2009). Consequently, in addition to fluctuations in $O_2$ and post-translational modification of PHD catalytic residues (Briggs et al, 2016; Lee et al, 2016), αKG levels may also determine the turnover kinetics of HIF1α. However, it is not known which of the pathways involved in αKG metabolism regulate HIF1α expression dynamics during the onset of hypoxia.

Upon its stabilisation in hypoxia, HIF1α controls the transcription of genes that include glucose transporters and most glycolytic genes (Semenza, 2013). Concomitantly, HIF1α drives the expression of pyruvate dehydrogenase kinase 1 (PDK1), which catalyses the inhibitory phosphorylation of pyruvate dehydrogenase (PDH), leading to attenuated pyruvate oxidation and, consequently, decreased contribution of glucose-derived carbons into the tricarboxylic acid (TCA) cycle (Kim et al, 2006; Papandreou et al, 2006). It has been postulated that decreased TCA cycle activity attenuates mitochondrial $NAD^+$-regenerating pathways, such as the malate-aspartate shuttle (MAS), leading to increased reliance of glycolysis on lactate dehydrogenase A (LDHA) for $NAD^+$ (Eales et al, 2016; Young and Anderson, 2008). Increased availability of pyruvate, the LDHA substrate, in the cytoplasm following PDH inhibition promotes LDHA activity (Wigfield et al, 2008). Moreover, the *LDHA* gene is also a HIF1α target, resulting in enhanced LDHA protein expression in hypoxia to further increase $NAD^+$ production (Locasale and Cantley, 2011). Accordingly, cells that rely more on glycolysis are more sensitive to inhibition of LDHA compared to cells that depend on mitochondria for ATP production (Boudreau et al, 2016). Furthermore, knock-down or pharmacological inhibition of LDHA in hypoxic cancer cells results in decreased proliferation and leads to cell death attributed to oxidative stress (Fantin et al, 2006; Le et al, 2010; Shim et al, 1997; Xie et al, 2009). Collectively, this evidence indicates that, increased LDHA protein expression, in addition to that of glucose transporters and glycolytic enzymes, is also required for increased glycolysis in chronic hypoxia (Hance et al, 1980; Zdralevic et al, 2017).

Intriguingly, while changes in gene expression through HIF1α, or other mechanisms, require more than 24 h to reach maximal levels (Lal et al, 2001), upregulation of glycolysis occurs within minutes to hours upon exposure to hypoxia and is essential for sustaining ATP levels under these conditions (Burgman et al, 2001; Clavo et al, 1995; Gerweck et al, 1993; Mertens et al, 1990). Acute increase in glycolysis upon hypoxia, or after inhibition of mitochondrial respiration, is due to the reversal of the Pasteur effect, which describes the inhibitory effect of oxygen on glycolysis. The Pasteur effect is mediated, in part, by increased activity of phosphofructokinase (PFK) due to decreased production of its allosteric inhibitor ATP in mitochondria (Krebs, 1972; Passonneau and Lowry, 1962), and increased phosphorylation by adenosine

monophosphate-activated protein kinase (AMPK) (Hardie, 2000; Marsin et al, 2002). Moreover, oxygen limitation can also upregulate glycolysis by directly influencing glucose uptake through mechanisms that include modification of glucose transporters or their increased translocation to the plasma membrane (Barros et al, 2007; Burgman et al, 2001; Clavo et al, 1995; Liemburg-Apers et al, 2016; Morgan et al, 1961; Shetty et al, 1993). Although our understanding of the processes involved in the initiation of the Pasteur effect is substantial, the mechanisms used to provide enough $NAD^+$ to support the upregulation of glycolysis during the onset of hypoxia remain elusive. In particular, in light of the apparent need for increased expression of LDHA in chronic hypoxia, it is unclear whether basal LDHA expression suffices to sustain redox balance also in early hypoxia, prior to HIF1α-mediated effects, or whether other mechanisms exist to support the cellular requirements for $NAD^+$ upon acute oxygen limitation.

Here we show that glycolysis increases within 3 h of exposure to hypoxia in a HIF1α-independent manner. Reserve LDHA capacity provides additional $NAD^+$ that, however, is not sufficient to sustain a maximal increase of glycolysis in early hypoxia, as evidenced by efflux of glucose carbons to α-glycerophosphate. Because of this limitation, maintenance of malate dehydrogenase 1 (MDH1) activity by GOT1 (glutamate-oxoglutarate transaminase 1, also known as aspartate aminotransferase 1), becomes more important in hypoxia than in normoxia for ATP homeostasis and cellular survival. In addition, GOT1 consumes αKG leading to attenuated PHD activity and increased HIF1α stabilisation in early hypoxia, and robust HIF1α target gene expression in later hypoxia.

## Results

### Glycolysis increases within 3 h in hypoxia and correlates with decreased aspartate levels

To investigate metabolic changes elicited by low oxygen concentrations, we measured intracellular metabolites in MCF7 cells incubated in 21% (normoxia) or 1% $O_2$ (hypoxia) for increasing lengths of time, between 1 and 24 h (Fig. 1A). The earliest and most statistically significant changes we observed were an increase in lactate and a decrease in intracellular aspartate levels, both of which persisted into later time points (Fig. 1B–D). These changes also occurred, to varying degrees, in other breast cancer cell lines, as well as immortalised, non-tumorigenic mammary epithelial cells (Appendix Fig. S1A,B). Upon reoxygenation following 3 h in hypoxia, lactate and aspartate levels in MCF7 cells returned to pre-hypoxia levels with comparable, albeit slower, kinetics than their onset, indicating that hypoxia-induced changes in lactate and aspartate are reversible (Appendix Fig. S1C,D). Treatment of cells with antioxidants did not attenuate the increase in lactate or decrease in aspartate (Appendix Fig. S1E–G), negating an involvement of increased reactive oxygen species (ROS), which have other important signalling roles in early hypoxia (Chandel et al, 1998; Guzy et al, 2005). Accumulation of intracellular lactate coincided with increased cellular glucose uptake at 1% $O_2$ [(1.69 ± 0.02)-fold and (2.78 ± 0.78)-fold after 3 h and 24 h, respectively] (Fig. 1E) and was accompanied by elevated lactate excretion into the media (Fig. 1F). Isotopic labelling with [U-$^{13}$C]-glucose showed that $^{13}$C-labelling of secreted lactate was higher in

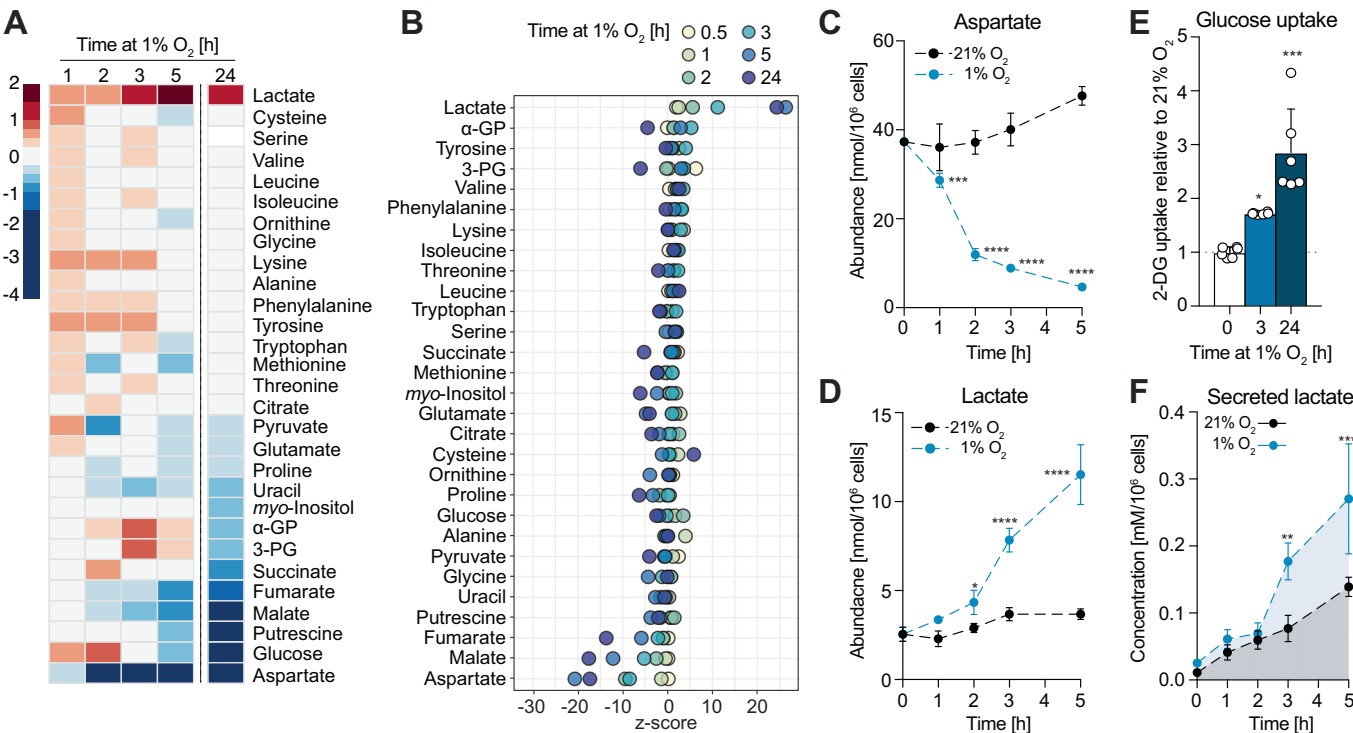

**Figure 1. Increased glycolysis occurs within 3 h upon exposure to 1% O₂ and correlates with decreased intracellular aspartate levels.**

(A) Heatmap showing log₂ fold changes of the abundance of the indicated metabolites in MCF7 cells exposed to 1% O₂ for the indicated lengths of time, compared to cells at 21% O₂. Metabolites are ordered according to log₂ fold changes after 24 h in 1% O₂. (B) Z-score plot of changes in metabolite abundances shown in (A). Metabolites are ordered according to their z-score values at 3 h in 1% O₂. (C, D) Intracellular abundances of aspartate and lactate, respectively, shown in (A). See also Appendix Fig. S1A,B. (E) Glucose (2DG) uptake of MCF7 cells in 21% O₂ and after 3 or 24 h in 1% O₂. (F) Lactate concentration in culture media of MCF7 cells incubated in 21% O₂ or 1% O₂ for the indicated lengths of time. Data information: Data are representative of experiments with similar conditions performed independently N times as follows: N ≥10 (A–D, 3 h), N ≥2 (A–D other time points and E, F). Datapoints in (C, D) represent mean ± s.d. n = 4 (A–D, F) and n = 6 (E) cultures per time point and condition, except t = 0 in (F) (n = 1), which corresponds to media without cells. P values for differences between 21% vs 1% O₂ were calculated by two-way ANOVA Sidak's test (C, D, F) or one-way ANOVA Dunnett's test (E). *P < 0.05, **P < 0.01, ***P < 0.001, ****P < 0.0001. Source data are available online for this figure.

hypoxia (Appendix Fig. S1H), which, together with the increased total abundance indicate that increased lactate is due to enhanced glycolysis. Together, these data showed that increased glycolysis occurs within 3 h after cells are exposed to 1% O₂ and coincides with decreased aspartate levels.

## Early increase in glycolysis is not dependent on HIF1α

Upregulation of glycolysis in chronic hypoxia is commonly attributed to the transcriptional activity of HIF1α, which results in increased glucose uptake and lactate production (Nakazawa et al, 2016). We found that HIF1α protein levels increased within 1 h and reached maximal levels within 3 h in hypoxia in all cell lines tested (Fig. 2A left; Appendix Fig. S2A). Expression of HIF2α, which also has important roles in the regulation of gene expression in hypoxia (Keith et al, 2011), did not change detectably within the time frame tested. Although changes in mRNA expression of many known HIF1α target genes (Benita et al, 2009) were detected in MCF7 cells after 3 h at 1% O₂, transcriptional up- and downregulation of most genes within this panel was more pronounced after 24 h in hypoxia (Fig. 2B). Despite the early onset of the transcriptional response (within 3 h in hypoxia), changes in protein levels of HIF1α targets involved in glucose uptake (GLUT1), glycolysis (HK2, PKM2) and

lactate production (LDHA) were only detected after 6 h, but not after 3 h in hypoxia (Fig. 2A, left). Therefore, the early metabolic changes described above occurred before robust expression of HIF1α target genes involved in glycolysis and lactate production was detectable on the protein level.

To confirm that the early upregulation of glycolysis was independent of HIF1α transcriptional activity, we engineered MCF7 cells that lack functional HIF1α (henceforth referred to as HIF1α^mut cells) using CRISPR/Cas9 (Appendix Fig. S2B). HIF1α^mut cells showed a severe impairment in upregulating HIF1α target genes in hypoxia, both on the mRNA and protein level (Fig. 2A right, C). Moreover, decreased entry of glucose carbons into the TCA cycle, which occurs in a HIF1α-dependent manner in chronic hypoxia (Papandreou et al, 2006), was partially attenuated in HIF1α^mut cells after 24 h in 1% O₂ (Fig. 2D; Appendix Fig. S2C); this attenuation was detectable even though suppression of glucose labelling into the TCA in wt cells is likely masked by continuous labelling throughout the incubation in hypoxia. Similarly, lactate accumulation at later time points (6 and 24 h in 1% O₂), was partially suppressed in HIF1α^mut cells compared to wild-type MCF7 cells (Fig. 2E). However, within the first 3 h in hypoxia, HIF1α^mut cells showed similar increases in lactate and decreases in aspartate as wild-type MCF7 cells (Fig. 2E). Together, these data

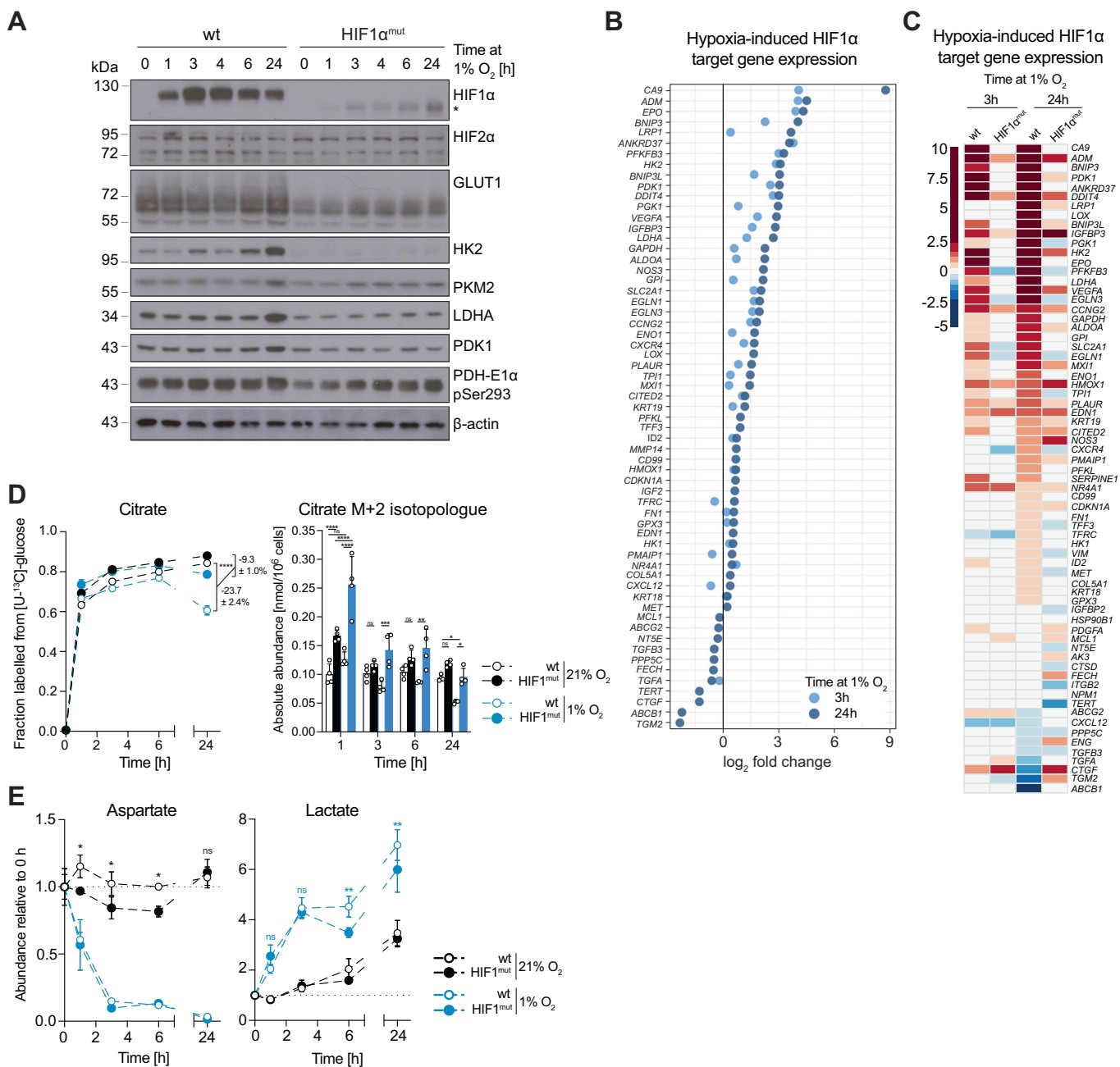

**Figure 2. Increased glycolysis and depletion of aspartate in early hypoxia are independent of HIF1α.**

(A) Western blot to assess levels of HIF1α and a panel of HIF1α targets in wild-type (wt) MCF7 and HIF1α$^{mut}$ MCF7 cells incubated in 21% $O_2$ or at 1% $O_2$ for the indicated lengths of time. The asterisk marks a HIF1α immunoreactive band of smaller molecular weight than HIF1α that increases upon hypoxia in HIF1α$^{mut}$ MCF7 cells, indicative of a truncated HIF1α that likely lacks the transactivation domain where the sgRNA sequence is targeted at. See also Appendix Fig. S2A. (B) Log$_2$ fold changes in mRNA expression levels of a panel of HIF1α targets in MCF7 cells exposed to 1% $O_2$ for 3 or 24 h, compared to control cells at 21% $O_2$. (C) Heatmap showing log$_2$ fold changes in mRNA expression levels of a panel of HIF1α targets in wild-type (wt) and HIF1α$^{mut}$ MCF7 cells exposed to 1% $O_2$ for 3 or 24 h, compared to control cells in normoxia. (D) Fraction labelled (left) and absolute abundances of the M + 2 isotopologue (right) of citrate from [U-$^{13}$C]-glucose in wild-type (wt) and HIF1α$^{mut}$ MCF7 cells after incubation with the tracer at 21% $O_2$ or 1% $O_2$ for the indicated lengths of time. Time points indicate both the duration of hypoxia treatment and incubation with the tracer. See also Appendix Fig. S2C. (E) Changes in lactate and aspartate abundance in wild-type (wt) and HIF1α$^{mut}$ MCF7 cells incubated in 21% $O_2$ or 1% $O_2$ for the indicated lengths of time, compared to control cells in normoxia. Data information: Data are representative of experiments with similar conditions performed independently N times as follows: $N \geq 2$ (A, D, E), $N = 1$ (B, C). Datapoints in (D, E) represent mean ± s.d. $n = 3$ (B, C) and $n = 4$ (D, E) cultures for each time point and condition. Statistical errors in (D, left and E) were propagated to calculate variance of the change in isotopic labelling between normoxia and hypoxia for each cell line. FDRs in (B, C) were calculated using the 'exactTest' function of the edgeR package (see 'Methods') with a cut-off set at 1%; only changes with FDR < 0.01 are shown. The P values shown were calculated by two-way ANOVA Sidak's test (D, left and E) or two-way ANOVA Tukey's test (D, right). ns non-significant, *P < 0.05, **P < 0.01, ***P < 0.001, ****P < 0.0001. Source data are available online for this figure.

demonstrated that, while later metabolic changes are, at least partially, dependent on HIF1α, the early increase in glycolysis upon hypoxia treatment occurs independently of HIF1α transcriptional activity. Henceforth, we refer to 3 h hypoxia as "early hypoxia" to distinguish it from other, previously described, acute responses mediated by ROS (Chandel et al, 1998; Guzy et al, 2005).

## Knock-out of GOT1 attenuates the increase in glycolysis in early hypoxia

To investigate mechanisms that sustain increased glycolysis in the absence of protein expression changes, we started by exploring further the strong counter-correlation between aspartate and lactate levels. We first asked whether decreased aspartate was due to decreased production or increased consumption. Incubation of cells with [U-$^{13}$C]-glucose or [U-$^{13}$C]-glutamine showed decreased labelling of aspartate from both labels at 1% O$_2$ vs 21% O$_2$ (Appendix Fig. S3A–D). Notably, within 5 h, labelling from glutamine had nearly reached isotopic steady state, whereas labelling from glucose had not. With this caveat in mind, we noted that fractional labelling of glutamate decreases from [U-$^{13}$C]-glutamine and increases from [U-$^{13}$C]-glucose, suggesting potential efflux of intermediates out of the TCA cycle. Incubation with supraphysiological concentrations (1.5 mM) of [U-$^{13}$C]-aspartate revealed a non-significant trend for decreased amount of labelled intracellular $^{13}$C-Asp, suggesting no substantial increase in aspartate consumption within 3 h in 1% O$_2$ vs 21% O$_2$ (Appendix Fig. S3E); in contrast, we observed a vast decrease in unlabelled aspartate. Together, these labelling data suggested that, within the timeframe tested, decreased production is a significant contributor to the low aspartate levels in early hypoxia.

Lactate accumulation accelerated after 2 h, when aspartate had decreased >75% compared to normoxic cells (Fig. 1C,D), raising the possibility that low aspartate levels may be required for the increase in lactate. To test this idea, we attempted to boost aspartate levels in hypoxia by providing cells with exogenous aspartate or its cell-permeable analogue, dimethyl-aspartate (DM-aspartate). Exogenous aspartate raised intracellular aspartate levels only modestly in normoxia, likely due to low expression of aspartate transporters (Garcia-Bermudez et al, 2018) in MCF7 cells, and had minimal effects on intracellular aspartate and lactate concentrations when cells were exposed to 1% O$_2$ for 3 h (Appendix Fig. S3F). The reason DM-aspartate failed to have a more pronounced effect on intracellular aspartate levels is less clear. To address the experimental limitation arising from poor exogenous aspartate uptake, and because at the time of these experiments, a specific aspartate transporter had not been identified, we cultured cells chronically with aspartate in the media and obtained an MCF7 derivative cell line that we named MCF7$^{Asp}$. In contrast to parental MCF7 cells, exogenous aspartate attenuated the hypoxia-induced decrease in intracellular aspartate in MCF7$^{Asp}$ cells without affecting the degree of increase in lactate (Appendix Fig. S3G). We therefore concluded that low aspartate levels are not causal to the increased lactate levels.

Aspartate is a substrate for GOT1, one of the enzymes that form the malate-aspartate shuttle (MAS), which links glycolysis and mitochondrial metabolism by transporting electrons across the inner mitochondrial membrane. Decreased aspartate levels in early hypoxia could curtail MAS activity, so we embarked on

investigating whether GOT1 is necessary for glycolysis in early hypoxia and, if yes, how it copes with decreased substrate availability. CRISPR/Cas9-mediated knock-out of GOT1 in MCF7 cells—henceforth referred to as GOT1ko cells—resulted in a 338 ± 45% increase in the steady-state levels of the intracellular aspartate pool (Fig. 3A,B). Upon hypoxia treatment, the decrease in aspartate persisted in GOT1ko cells (Fig. 3B). Concomitantly, glutamine-labelled aspartate increased more in hypoxia compared to normoxia in GOT1ko cells (Appendix Fig. S3H). We also observed a non-significant trend for increased malate labelling from [U-$^{13}$C]-aspartate (Appendix Fig. S3E). Together, these data suggested that GOT1 activity contributes to but does not, alone, account for the hypoxia-induced decrease in aspartate in wild-type cells. Interestingly, after 3 h in hypoxia, the upregulation of glucose uptake was only modestly attenuated, whereas the accumulation of both intracellular and secreted lactate were significantly suppressed in GOT1ko cells (Fig. 3C–E). Glucose uptake became significantly attenuated after 24 h in hypoxia in GOT1ko cells compared to wild-type cells (Fig. 3C). Ectopic expression of HA-tagged GOT1 reversed the accumulation of aspartate in GOT1ko cells and restored hypoxia-induced lactate to levels similar to those in wild-type MCF7 cells (Fig. 3F–H). These data suggested that, even though its substrate aspartate decreases, GOT1 activity is required to sustain the increase in glycolysis in early hypoxia.

## GOT1 contributes to cytoplasmic NAD$^+$/NADH balance by sustaining flux through MDH1

To further probe the requirement of GOT1 for glycolysis, we quantified glycolytic and pentose phosphate pathway (PPP) intermediates by liquid chromatography-mass spectrometry (LC-MS). Metabolite pools upstream of GAPDH increased, while downstream metabolites decreased in GOT1ko compared to wild-type cells (Fig. 4A; Appendix Fig. S4A). This metabolic profile indicated a bottleneck for glycolytic flux at GAPDH (Kornberg et al, 2018; Liberti et al, 2017), which was unlikely due to carbon substrate limitation since glucose uptake in normoxia was similar in both cell lines (Fig. 3C). GAPDH activity depends on the availability of NAD$^+$ and is attenuated by increased levels of NADH through competitive product inhibition (Aithal et al, 1985; Copeland and Zammit, 1994). We therefore quantified NAD$^+$ and NADH by LC-MS and compared the respective NAD$^+$/NADH ratios in wild-type and GOT1ko cells. In wild-type cells, 3 h in 1% O$_2$ led to a decrease in NAD$^+$/NADH ratio from 6.0 ± 0.2 to 3.4 ± 0.2 (Fig. 4B). In contrast, the NAD$^+$/NADH ratio in GOT1ko cells was already lower (3.6 ± 0.4) at 21% O$_2$ and did not change significantly upon incubation of cells in 1% O$_2$. Notably, comparable NAD$^+$/NADH ratios did not reflect equivalent redox states, because the hypoxia-induced decrease in NAD$^+$/NADH ratio in wild-type cells was due to increased NADH levels, while the lower NAD$^+$/NADH ratio in GOT1 cells reflected decreased NAD$^+$ levels (Fig. 4B–D). These findings are consistent with the idea that upregulation of glycolysis, associated with higher consumption of NAD$^+$ by GAPDH, imposes an increased need for NADH oxidation to maintain redox balance. They also indicate that regeneration of NAD$^+$ is compromised and may underlie impaired glycolysis in GOT1ko cells during early hypoxia.

The cellular pools of NAD(H) are asymmetrically distributed between subcellular compartments, due to the differential

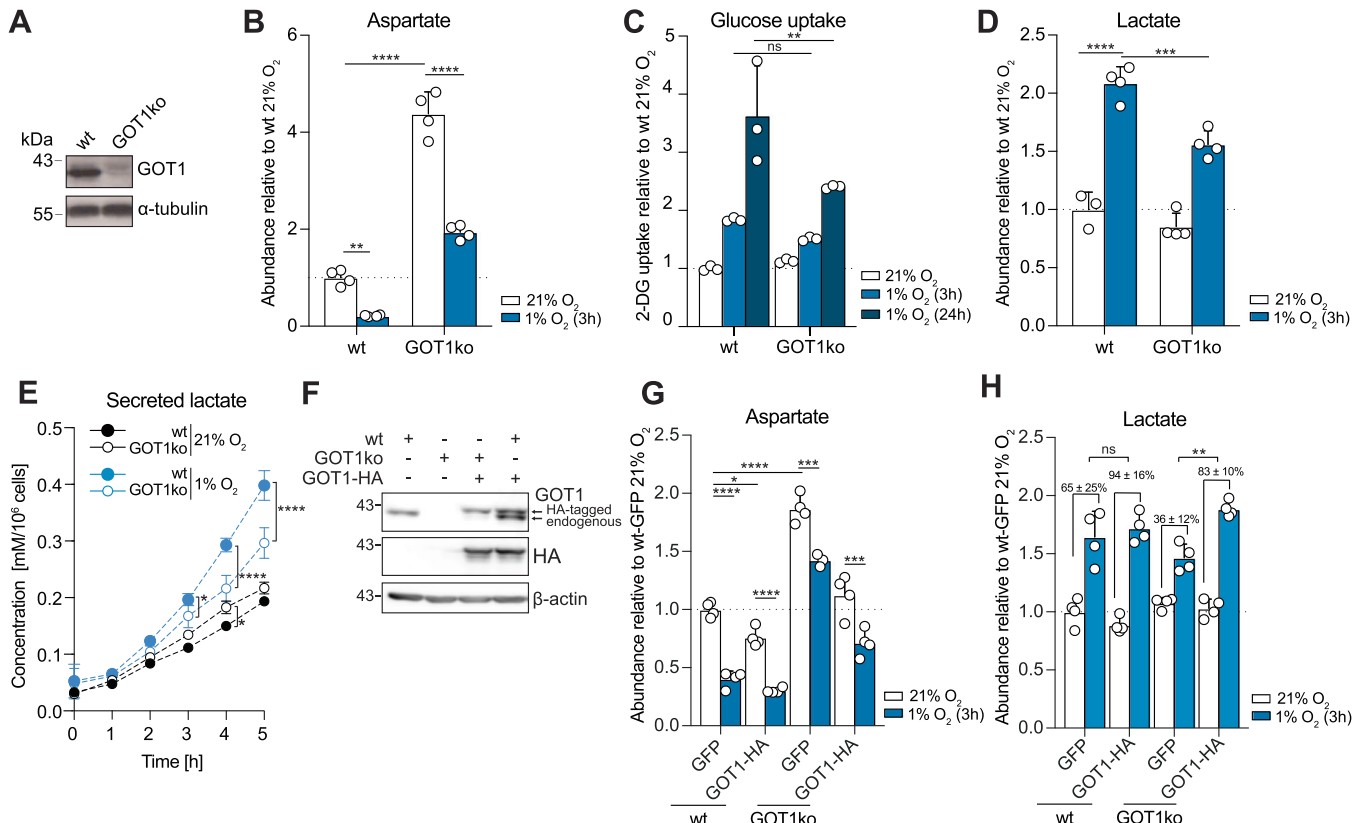

**Figure 3. GOT1 supports increased glycolysis in early hypoxia.**

(A) Western blot to assess levels of GOT1 in wild-type (wt) and GOT1ko MCF7 cells. (B) The intracellular abundance of aspartate in wild-type (wt) and GOT1ko MCF7 cells incubated in 21% $O_2$ or 1% $O_2$ for 3 h. (C) Glucose (2DG) uptake of wild-type (wt) and GOT1ko MCF7 cells in normoxia and after 3 and 24 h in 1% $O_2$. (D) The intracellular abundance of lactate in wild-type (wt) and GOT1ko MCF7 cells incubated in 21% $O_2$ or 1% $O_2$ for 3 h. (E) Lactate concentration in cell culture media of wild-type (wt) and GOT1ko MCF7 cells incubated in 21% $O_2$ or 1% $O_2$ for the indicated lengths of time. (F) Western blot to assess the levels of HIF1α, endogenous GOT1 and HA-tagged GOT1 in wild-type (wt) and GOT1ko MCF7 cells stably expressing GOT1-HA or GFP. (G, H) Intracellular abundance of aspartate and lactate in wild-type (wt) and GOT1ko MCF7 cells stably expressing GOT1-HA or GFP at 21% $O_2$ and after 3 h in 1% $O_2$, relative to wild-type cells at 21% $O_2$. Data information: Data are representative of experiments with similar conditions performed independently $N$ times as follows: $N \geq 9$ (A), $N \geq 5$ (B, D), $N \geq 2$ (C, E, G, H), $N \geq 3$ (F). Datapoints in (B–E, G, H) represent mean ± s.d. $n = 3$ assays per condition (C) and $n = 4$ cultures for each time point and condition (B, D, E, G, H), except for E 1% $O_2$, $t = 0$ and all measurements at 21% $O_2$ where $n = 3$. $P$ values shown were calculated by two-way ANOVA Sidak's test (C) or two-way ANOVA Tukey's test (B, D, E). Statistical errors in (H) were propagated to calculate the error of the change in lactate between normoxia and hypoxia for each condition, and significance between these changes was then tested using one-way ANOVA Tukey's test. ns non-significant, *$P < 0.05$, **$P < 0.01$, ***$P < 0.001$, ****$P < 0.0001$. Source data are available online for this figure.

localisation of pyridine nucleotide precursors and biosynthetic pathways, as well as the impermeability of the inner mitochondrial membrane to NADH and the low apparent affinity of a recently identified mitochondrial transporter for $NAD^+$ (Kory et al, 2020; Xiao et al, 2018). Since GAPDH activity depends on cytoplasmic $NAD^+/NADH$, we specifically assessed cytoplasmic redox state in wild-type and GOT1ko cells using the genetically encoded NADH sensor Peredox (Hung et al, 2011). To this end, we recorded the basal Peredox T-sapphire signal of individual cells incubated in buffer supplemented with regular concentrations of the main carbon sources glucose and glutamine, and subsequently compared it to the T-sapphire signal after sequential incubation with only 10 mM lactate or 10 mM pyruvate. In the absence of extracellular glucose to counter-balance cytoplasmic redox changes, incubation of cells with lactate leads to the production of NADH via LDHA and results in maximal Peredox T-sapphire signal that depends on the amount of available $NAD^+$ (Bucher et al, 1972; Hung et al, 2011). Conversely, incubation of cells with pyruvate leads to the

consumption of available cytoplasmic NADH via LDHA and minimises Peredox T-sapphire signal (Appendix Fig. S4B). Assessment of the basal Peredox T-sapphire, together with the maximal availability of NADH and $NAD^+$ reported by incubation with pyruvate and lactate, respectively, allows comparison of the cytoplasmic redox state in wild-type versus GOT1ko cells.

In wild-type MCF7 cells, the basal Peredox signal was similar to the high $NAD^+/NADH$ state (Pyr), while lactate treatment (Lac) caused a dramatic increase in Peredox signal intensity irrespective of the order of substrate addition (Fig. 4E; Appendix Fig. S4C). Conversely, in GOT1ko cells extracellular lactate failed to increase cytoplasmic NADH, further supporting our interpretation of the cell population-level $NAD^+/NADH$ quantification by LC-MS that GOT1ko cells have a deficit in cytoplasmic $NAD^+$. Accordingly, overexpression of GOT1-HA in wt cells enhanced the lactate-induced and suppressed the pyruvate-induced Peredox response; overexpression of GOT1-HA in GOT1ko cells restored the Peredox response to wt cell levels (Appendix Fig. S4D). Together, these

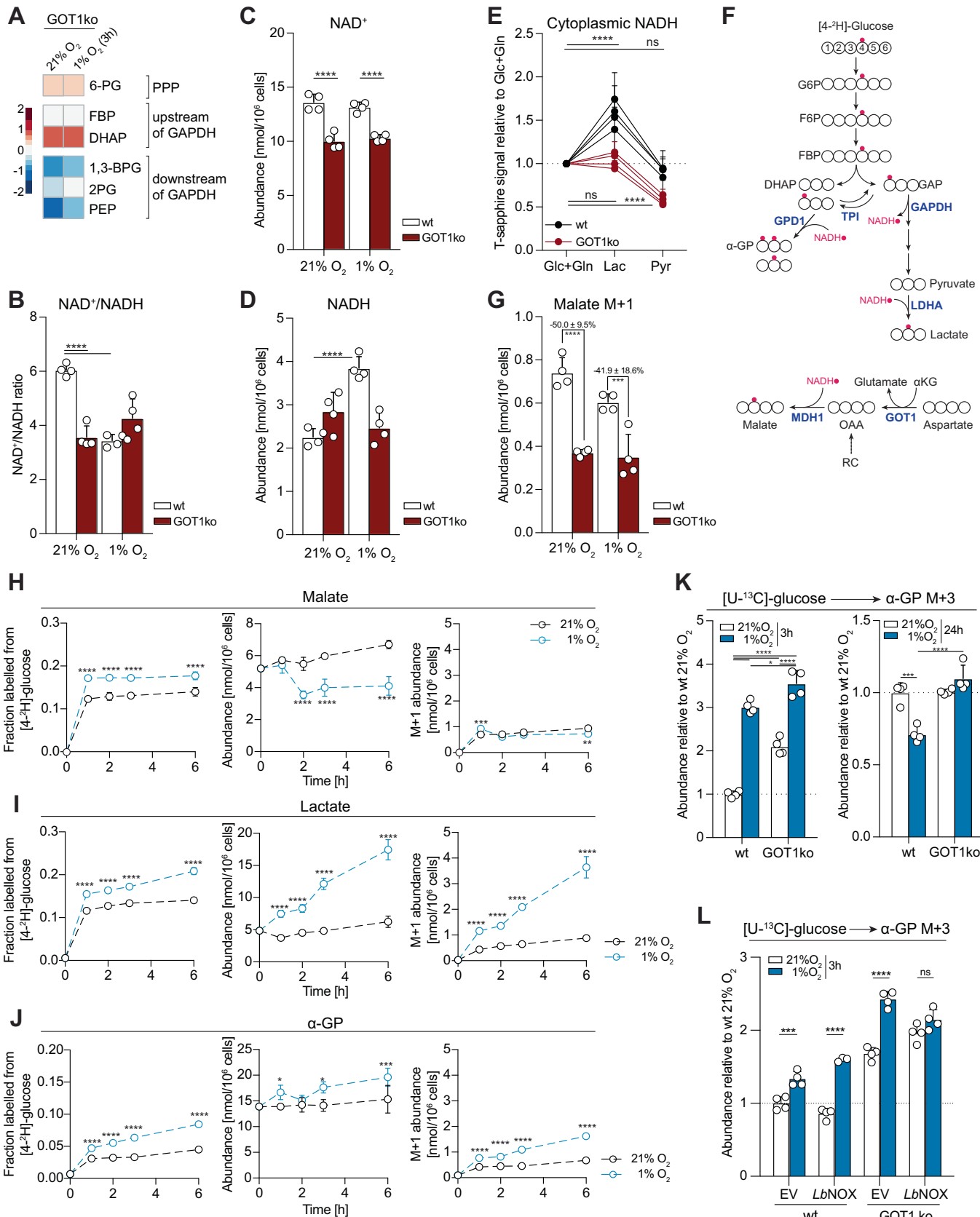

**Figure 4.   GOT1 supports MDH1 flux and cytoplasmic redox balance, but MDH1 flux does not change in hypoxia *vs.* normoxia.**

(A) Heatmap showing $\log_2$ fold changes in the abundance of the indicated metabolites in GOT1ko at 21% $O_2$ or after 3 h in 1% $O_2$, compared to wild-type MCF7 cells under the same conditions. Data for each condition separately are shown in Appendix Fig. S4A. 6-PG 6-phosphogluconic acid, FBP fructose 1,6-biphosphate, DHAP dihydroxyacetone phosphate, 1,3-BPG 1,3-biphosphoglyceric acid, 2PG 2-phosphoglyceric acid, PEP phosphoenolpyruvate. (B–D) $NAD^+$/NADH ratio (B) calculated from the intracellular abundance of $NAD^+$ (C) and NADH (D) in wild-type (wt) and GOT1ko MCF7 cells incubated in 21% $O_2$ or 1% $O_2$ for 3 h. (E) Peredox T-sapphire fluorescence signal intensity of wild-type (wt) and GOT1ko MCF7 cells in buffer containing 5.5 mM glucose and 2 mM glutamine (Glc+Gln) and after sequential incubation first with 10 mM lactate (Lac) and then with 10 mM pyruvate (Pyr). Signal was normalised per nucleus and is shown relative to the Glc+Gln condition. See also Appendix Fig. S4B,C. (F) Schematic showing theoretical labelling patterns in the indicated metabolites from [4-$^2$H]-glucose. Carbon atoms are shown in white and deuterium atoms are shown in red. Adapted from (Lewis et al, 2014). (G) The intracellular abundance of the M + 1 isotopologues of malate in wild-type (wt) and GOT1ko MCF7 cells after incubation with [4-$^2$H]-glucose for 3 h. See also Appendix Fig. S4E. (H–J) Fraction labelled from [4-$^2$H]-glucose, absolute total abundances and absolute abundances of M + 1-labelled isotopologues from [4-$^2$H]-glucose of the shown metabolites. Time points indicate duration of incubation at 21% $O_2$ or 1% $O_2$, as well as duration of incubation with the isotopic tracer. (K) The intracellular abundance of α-glycerophosphate (α-GP) M + 3 labelled from [U-$^{13}$C]-glucose in wild-type (wt) and GOT1ko MCF7 cells after the indicated lengths of time in 21% $O_2$ or 1% $O_2$. Cells were incubated with the tracer for 3 or 24 h, respectively. (L) Intracellular abundance of α-glycerophosphate (α-GP) M + 3 labelled from [U-$^{13}$C]-glucose in wild-type (wt) and GOT1ko MCF7 cells, stably expressing an empty vector (EV) or *Lb*NOX. Cells were incubated with the tracer for 3 h in 21% $O_2$ or 1% $O_2$, respectively. Data information: Data are representative of experiments with similar conditions performed independently *N* times as follows: $N \geq 2$ (A, K), $N \geq 3$ (B–D, G), $N = 4$ (E), $N = 1$ (H–J, except 3 h time-point where $N = 3$, and L). Datapoints in (B–E, G–L) represent mean ± s.d. $n = 4$ cultures for each time point or cell line and condition (A–D, G, H–L) except for (H–J): 1% $O_2$, 2 h, $n = 3$ and 21% $O_2$, 3 h, $n = 2$. Data points in (E) represent mean ± s.d. of four independent replicates per cell line ($n = 25$–55 cells per replicate). *P* values shown were calculated by two-way ANOVA Dunnett's test (B–E), two-way ANOVA Sidak's test (G–J, L) or two-way ANOVA Tukey's test (K). ns non-significant, *$P < 0.05$, **$P < 0.01$, ***$P < 0.001$, ****$P < 0.0001$. Source data are available online for this figure.

observations indicated that the attenuated increase in glycolysis of GOT1ko cells in early hypoxia may be due to decreased $NAD^+$ availability.

GOT1 converts aspartate to oxaloacetate (OAA), a substrate of MDH1, which produces malate and concomitantly oxidises NADH to $NAD^+$ (Fig. 4F). To assess whether deletion of GOT1 influences MDH1 activity and thereby the production of cytoplasmic $NAD^+$, we incubated cells with [4-$^2$H]-glucose, which leads to the production of cytoplasmic NAD$^2$H that can be subsequently used by MDH1 to incorporate a deuterium into malate (malate M + 1) (Lewis et al, 2014) (Fig. 4F). Malate M + 1 abundance was 50 ± 10% and 42 ± 19% lower in normoxia and early hypoxia, respectively, in GOT1ko compared to wild-type cells (Fig. 4G), while labelling of NADH was similar in both cell lines and conditions (Appendix Fig. S4E). These results show that a significant fraction of MDH1 flux depends on GOT1 both in normoxia and in early hypoxia, consistent with the lower basal $NAD^+$ levels we observed in GOT1ko cells (Fig. 4C).

## MDH1 flux does not increase in early hypoxia

Other reports have previously indicated that increased flux through MDH1 is required to support glycolysis (Gaude et al, 2018) and this may be associated with increased MDH1 expression (Hanse et al, 2017). We also observed an increase in MDH1, but not GOT1 protein levels in hypoxia (Appendix Fig. S4F), however, we found no significant differences in malate M + 1 levels between normoxia and hypoxia in either cell line (Fig. 4G). We therefore explored the contribution of GOT1, *versus* other potential OAA sources, to MDH1 flux in early hypoxia.

In wild-type MCF7 cells, the incorporation of deuterium into malate reached steady state within 1 h in hypoxia (Fig. 4H, left panel). After accounting for the significant decrease in the malate pool size (Fig. 4H, middle panel), we found that the abundance of malate M + 1 at steady-state was similar in 21% $O_2$ and 1% $O_2$ (Fig. 4H, right panel). Malate M + 1 levels remained relatively constant even after 6 h in hypoxia, despite the progressive decrease in aspartate over time (Appendix Fig. S4G), indicating that aspartate did not become limiting for GOT1-MDH1 flux.

Furthermore, overexpression of ectopic GOT1-HA in wild-type MCF7 cells modestly enhanced lactate accumulation (Fig. 3H), suggesting that lower glycolysis under these conditions may be limited by GOT1 expression.

Reductive carboxylation (RC) has also been proposed as a source of OAA for MDH1 to support glycolysis in cells with mitochondrial defects or upon growth factor stimulation (Gaude et al, 2018; Hanse et al, 2017). Similar to previous reports (Metallo et al, 2011), RC increased in MCF7 cells after 24 h in hypoxia, however, we found no significant increase in RC after 3 h in hypoxia (Appendix Fig. S4H). GOT1ko cells showed a modest increase in RC (Appendix Fig. S4I), but this was not sufficient to fully restore MDH1 flux (Fig. 4G) and prevent the observed attenuation of glycolysis in early hypoxia (Fig. 3C,D).

Together, these data showed that MDH1 flux is maximal in normoxia and does not further increase in early hypoxia.

## $NAD^+$ is limiting for the maximal flow of carbons from upper to lower glycolysis in early hypoxia

Given the lack of increase in MDH1 flux during early hypoxia, we investigated whether LDHA, which has an established role in supporting increased glycolysis in chronic hypoxia (Fantin et al, 2006; Le et al, 2010) has a similar role also in early hypoxia. In normoxia, the amount of lactate M + 1 produced from [4-$^2$H]-glucose was comparable to that of malate M + 1 (Fig. 4H, I, right panels). In hypoxia, incorporation of $^2$H into lactate increased linearly over time, indicating an increased contribution of LDHA to $NAD^+$ production (Fig. 4I). We concluded that, in contrast to MDH1, flux through LDHA in normoxia is not maximal and can increase in early hypoxia despite the lack of increased LDHA protein expression.

Interestingly, we observed that the production of α-glycerophosphate (α-GP, glycerol 3-phosphate) M + 1 from [4-$^2$H]-glucose also increased in hypoxia (Fig. 4J), which was further reflected by the increased labelling of α-GP from [U-$^{13}$C]-glucose (Fig. 4K, wt cells on the left). α-GP is produced by α-GP dehydrogenase 1 (GPD1), from dihydroxyacetone phosphate (DHAP), which, alongside glyceraldehyde 3-phosphate (GAP) is a

product of aldolase (Appendix Fig. S4J). DHAP and GAP are interconverted by triose phosphate isomerase (TPI), which thermodynamically favours DHAP formation. However, in cells, high GAPDH activity rapidly consumes GAP and shifts the TPI equilibrium towards GAP formation, thereby allowing the reactions of lower glycolysis to occur (Aithal et al, 1985; Amelunxen and Grisolia, 1962; Harris et al, 1998; Herlihy et al, 1976; Tucker and Grisolia, 1962; Veech et al, 1969). Therefore, increased incorporation of glucose carbons to α-GP may reflect a limitation in carbon flow from upper to lower glycolysis due to attenuated GAPDH activity, increased GPD1 activity, or both.

Consistent with this idea, GOT1ko cells, which have impaired NAD$^+$-regenerating capacity (Fig. 4G) and attenuated lower glycolysis (Fig. 4A), show increased labelling of α-GP from [U-$^{13}$C]-glucose compared to wild-type cells in normoxia (Fig. 4K, left) despite lower glucose uptake (Fig. 3C). Expression of the bacterial NADH oxidase *Lb*NOX (Titov et al, 2016) completely prevented the hypoxia-induced increase in α-GP labelling from [U-$^{13}$C]-glucose in GOT1ko cells but not in wild-type cells (Fig. 4L), further supporting our model that elevated efflux of glucose carbons to α-GP in early hypoxia when GOT1 is absent reflects limiting NAD$^+$. Expression of exogenous GOT1-HA in GOT1ko cells decreased α-GP to similar levels as those found in parental cells in hypoxia and partly reversed the impairment in MDH1 flux (Fig. S4K). Intriguingly, α-GP labelling decreased in wild-type cells after 24 h in hypoxia, compared to normoxia, but remained elevated in GOT1ko cells (Fig. 4K, right). Together, these data suggested that, in early hypoxia, cellular NAD$^+$-regenerating capacity is not sufficient for maximal flow of carbons from upper to lower glycolysis. After 24 h in hypoxia, decreased α-GP production indicates that flux through the reactions of lower glycolysis can match, or exceed, that of upper glycolysis, likely because of increased NAD$^+$-regenerating capacity due to higher LDHA protein expression (Fig. 2A) or increased RC via MDH1 (Appendix Fig. S4K).

We next tested whether LDHA is sufficient for sustaining lower glycolysis or whether its function can be substituted by GOT1-MDH1. Knock-out of LDHA (LDHAko) in MCF7 cells effectively abrogated the production of lactate from glucose and led to an accumulation of labelled pyruvate (Fig. 5A,B), demonstrating that LDHA is the predominant LDH isoform in these cells. α-GP labelling from glucose increased in LDHAko cells, pointing to a bottleneck between upper and lower glycolysis. This interpretation was confirmed by the observed increase in the levels of metabolites in upper glycolysis and depletion of metabolites in lower glycolysis both under normoxia and, to a greater extent, under hypoxia (Fig. 5C; Appendix Fig. S5A). Notably, these changes in glycolytic intermediates were more pronounced than those in GOT1ko cells (Fig. 4A; Appendix Fig. S4A). This observation, combined with the enhanced synthesis of α-GP from glucose in LDHAko versus GOT1ko cells, revealed a greater reliance of lower glycolysis on LDHA than GOT1-dependent MDH1 flux. Together, our data indicated that LDHA is necessary but not sufficient, even combined with GOT1-MDH1, to sustain maximal carbon flow from upper to lower glycolysis in hypoxia.

## GOT1 and LDHA synergistically maintain ATP homeostasis and cell survival in hypoxia

An important function of increased glycolysis in chronic hypoxia, when respiration is suppressed, is to maintain intracellular ATP

levels (Kroemer and Pouyssegur, 2008). We observed a comparable decrease in respiration of wt cells at 3 and 24 h hypoxia (23 ± 18% vs 37 ± 11%, respectively, Fig. 5D), that was accompanied by a more pronounced decrease in ATP after 24 h (31 ± 4%) than after 3 h in hypoxia (6 ± 2%, Fig. 5E). These results suggested that increased glycolysis could also preserve ATP levels upon suppression of mitochondrial respiration in early hypoxia.

As our data pointed to a differential role for GOT1-MDH1 and LDHA in sustaining lower glycolysis, we compared the relative ability of GOT1ko and LDHAko cells to maintain ATP homeostasis. ATP levels in both GOT1ko and LDHAko cells were comparable to those in wild-type cells in normoxia (Fig. 5E), suggesting that remaining flux through lower glycolysis after deletion of either GOT1 or LDHA is sufficient to maintain ATP homeostasis in normoxia. Under hypoxia, ATP levels decreased similarly in wild-type and GOT1ko cells, but more significantly in LDHAko cells, both at 3 h and 24 h (Fig. 5E). Expression of low levels of exogenous LDHA attenuated the hypoxia-induced decrease in ATP from 56% to 32% (Appendix Fig. S5B). ATP depletion in LDHAko cells was accompanied by a 68 ± 8% loss of cell mass after 24 h in hypoxia compared to normoxia, whereas, under the same conditions, wild-type cells showed a more modest decrease in cell mass (15 ± 4%, Appendix Fig. S5C). These data are in line with the increased reliance of lower glycolysis on LDHA compared to GOT1-MDH1 in early hypoxia and indicate that MDH1 fuelled by OAA from GOT1, or other sources, cannot compensate for decreased LDHA activity. Conversely, our results suggest that LDHA may suffice to support ATP production upon loss of GOT1 (Fig. 5F).

To test the dependence of GOT1ko cells on LDHA for maintaining ATP levels, we treated GOT1ko cells with oxamate, a competitive inhibitor of LDHA (Novoa et al, 1959) that led to a dose-dependent decrease in the production of lactate from [U-$^{13}$C]-glucose (Fig. 5G, left). In agreement with our findings in LDHAko cells, treatment with oxamate had no effect on ATP levels in wild-type cells in normoxia but led to a dose-dependent decrease in ATP levels in hypoxia (Fig. 5H), without affecting mitochondrial respiration (Appendix Fig. S5D). The oxamate-induced decrease in ATP under hypoxia correlated remarkably well ($R^2 = 0.975$) with the magnitude of the increase in α-GP synthesis from glucose under the same conditions (Fig. 5G, right, I), further supporting our model that increased α-GP synthesis reflects efflux of glucose carbons from glycolysis via GPD1 and diminished activity of lower glycolysis. In contrast to untreated GOT1ko cells, treatment of GOT1ko cells with oxamate resulted in lower ATP levels even in normoxia. This decrease in ATP was further exacerbated after 3 h in hypoxia (Fig. 5H) but did not occur either in normoxia or hypoxia with oligomycin (Appendix Fig. S5E). Furthermore, oxygen consumption in GOT1ko cells was modestly decreased compared to that of wild-type cells (Appendix Fig. S5F). Together, these observations indicated that GOT1ko cells rely more on glycolytic flux supported by LDHA, rather than compensatory mitochondrial respiration, to maintain intracellular ATP levels. Importantly, oxamate caused a more profound decrease in GOT1ko than wild-type cell number in hypoxia than in normoxia, and this decrease was rescued by exogenous GOT1-HA (Fig. 5J; Appendix Fig. S5G). Collectively, these data revealed that while GOT1-MDH1 and LDHA contribute differentially to ATP production from lower glycolysis, they synergise to maintain

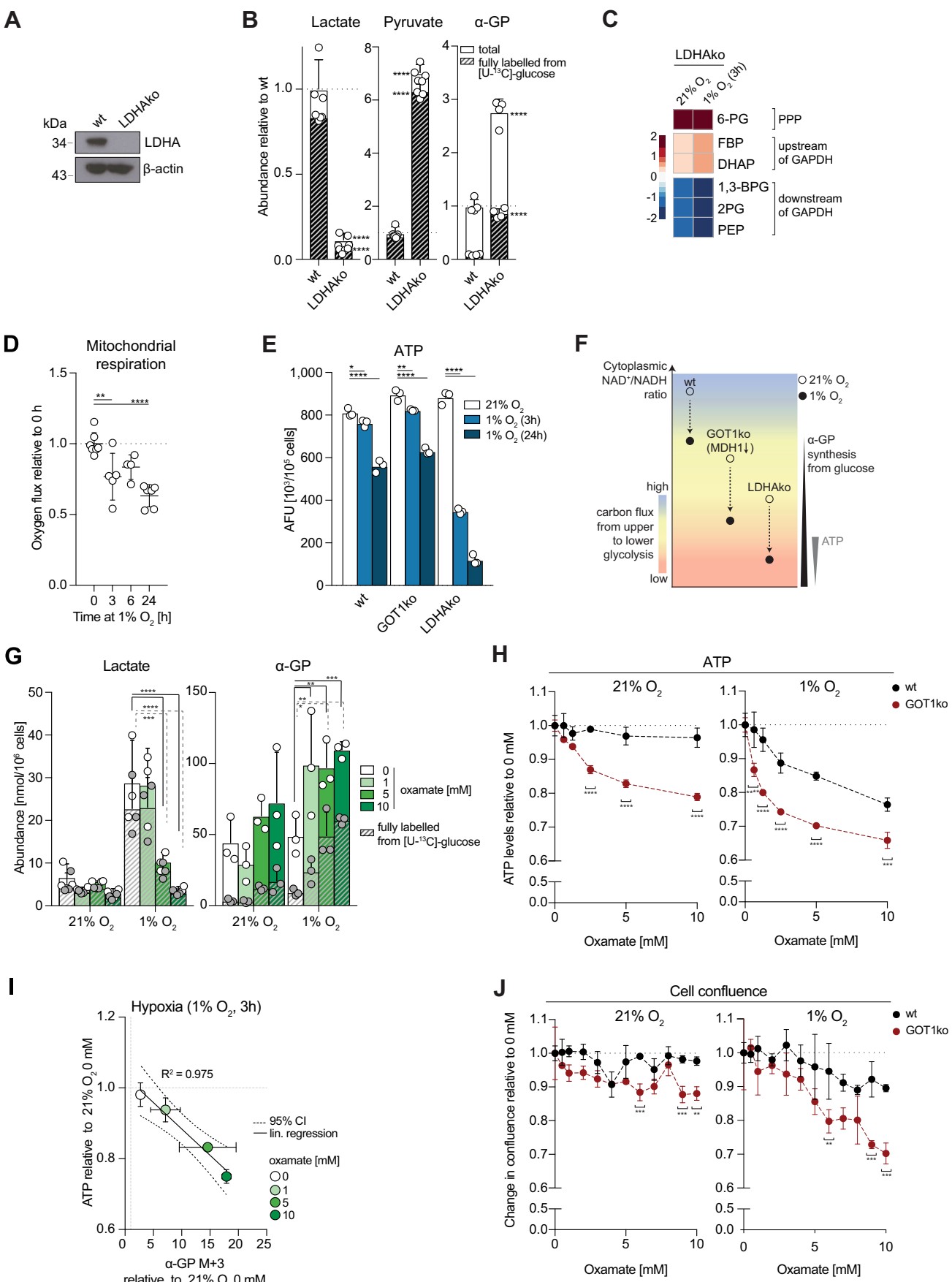

◄   **Figure 5.   LDHA has spare capacity in normoxia and is necessary to maintain ATP levels in early hypoxia.**

(A) Western blot to assess the levels of LDHA in wild-type (wt) and LDHAko MCF7 cells. (B) The intracellular abundance of pyruvate, lactate and α-glycerophosphate (α-GP) in wild-type (wt) and LDHAko MCF7 cells. Striped bars represent the fraction of metabolites fully labelled from [U-$^{13}$C]-glucose after 3 h incubation with the tracer. (C) Heatmap showing $\log_2$ fold changes in the abundance of the indicated metabolites in LDHAko at 21% $O_2$ or after 3 h in 1% $O_2$, compared to wild-type MCF7 cells in the same conditions. Data for each condition separately are shown in Appendix Fig. S5A. Wild-type data are the same as shown in Fig. 4A and Appendix Fig. S4A and statistical tests were performed on the whole data set. 6-PG 6-phosphogluconic acid, FBP fructose 1,6-biphosphate, DHAP dihydroxyacetone phosphate, 1,3-BPG 1,3-biphosphoglyceric acid, 2PG 2-phosphoglyceric acid, PEP phosphoenolpyruvate. (D) Mitochondrial respiration of MCF7 cells after incubation at 1% $O_2$ for the indicated lengths of time. Cellular oxygen consumption was corrected for ROX (residual oxygen consumption) by the addition of the complex III inhibitor antimycin A. (E) ATP levels in wild-type (wt), GOT1ko and LDHAko MCF7 cells at 21% $O_2$ and after 3 h or 24 h in 1% $O_2$. See also Appendix Fig. S5B. (F) Schematic of a theoretical working model, for illustrative purposes, summarising the observed effects of GOT1 and LDHA deletion (GOT1ko and LDHAko, respectively) on carbon flux from upper to lower glycolysis (indicated by the colour scale) and changes in cellular ATP during early hypoxia. Decreased NAD$^+$/NADH ratio in GOT1ko cells leads to an attenuation of carbon flux into lower glycolysis that is not large enough to affect ATP levels. In contrast, loss of LDHA leads to more profound inhibition of lower glycolysis, associated with ATP depletion and cell death. wt: wild-type cells. (G) Intracellular abundance of lactate and α-glycerophosphate (α-GP) in MCF7 cells treated with a range of concentrations of the LDHA inhibitor oxamate for 3 h at 21% $O_2$ or 1% $O_2$. Striped bars represent the fraction of metabolites fully labelled from [U-$^{13}$C]-glucose after 3 h incubation with the tracer. (H) ATP levels in wild-type (wt) and GOT1ko MCF7 cells at 21% $O_2$ and after 3 h in 1% $O_2$ treated with the indicated oxamate concentrations for 3 h. (I) Scatter plot showing changes in abundance of α-glycerophosphate (α-GP) M + 3 labelled from [U-$^{13}$C]-glucose (3 h incubation with the tracer) versus the corresponding changes in ATP levels, in MCF7 cells treated with a range of oxamate concentrations for 3 h at 1% $O_2$ relative to cells treated at 21% $O_2$. (J) Change in cell confluence of wild-type (wt) and GOT1ko MCF7 cells within 24 h at 21% $O_2$ or 1% $O_2$ with the indicated oxamate concentration in cell culture media, shown relative to 0 mM oxamate per cell line. Data information: Data are representative of experiments with similar conditions performed independently N times as follows: N = 2 (B, D, H, J), N ≥3 (A), N = 1 (C, E, G, I). Datapoints in (B, D, G–J) represent mean ± s.d. n = 4 (B, C) and n = 3 (E, I, J) cultures for each cell line and condition; n = 3 assays per cell line, time point and condition; n = 3 cultures per condition; (D) graphs show combined replicates of two independent experiments: [n = 7 (0 h); n = 5 (3 h); n = 4 (6 h); n = 6 (24 h)]. P values shown were calculated by two-way ANOVA Dunnett's test (D, E, G) or two-way ANOVA Sidak's test (B, H, J). ns non-significant, *P < 0.05, **P < 0.01, ***P < 0.001, ****P < 0.0001. Source data are available online for this figure.

ATP homeostasis in early hypoxia and contribute to cell survival after 24 h in hypoxia.

In summary, our results suggest that enhanced upper glycolysis in early hypoxia imposes an increased need for NAD$^+$ to sustain maximal flow of carbons to lower glycolysis. This increased requirement for NAD$^+$ is met by reserve LDHA capacity and is further supported by (saturated) GOT1-MDH1, to sustain ATP homeostasis in early hypoxia. However, even with the combined action of LDHA and MDH1, NAD$^+$ regeneration is not enough to achieve the maximal flow of carbons from upper to lower glycolysis.

## GOT1 consumes αKG to attenuate PHD activity and promote HIF1α stabilisation

To investigate whether loss of GOT1 also affects the long-term hypoxic response, we first monitored wild-type and GOT1ko cell proliferation over 2 days. We found that loss of GOT1 did not affect proliferation either in normoxia or hypoxia (Appendix Fig. S6A). We next compared gene expression in wild-type and GOT1ko cells incubated in 1% $O_2$ for 24 h. Both cell lines exhibited widespread gene expression changes in hypoxia, compared to normoxia, including changes in HIF1α target genes (Appendix Fig. S6B and Dataset EV1. However, the induction of HIF1α target gene expression was markedly suppressed in GOT1ko cells compared to wild-type cells (Fig. 6A). This suppression was not due to a defect in transcription, as the profile of global gene expression changes induced by hypoxia in GOT1ko cells was largely similar to that of wild-type cells (Appendix Fig. S6C). Therefore, these data show that HIF1α-dependent transcription is attenuated in GOT1ko cells.

Decreased induction of HIF1α target mRNAs was associated with both a delay and decrease in the hypoxia-induced accumulation of HIF1α protein in GOT1ko cells (Fig. 6B). This was particularly evident within the first hours in hypoxia, whereas the kinetics of the decrease in HIF1α protein levels at longer times (>15 h) under hypoxia (Lin et al, 2011) were similar. Re-expression of HA-tagged GOT1 restored HIF1α expression, showing a GOT1-

specific effect (Fig. 6C). These data indicate that GOT1 promotes HIF1α stabilisation in early hypoxia and suggest that attenuation of HIF1α target gene expression in GOT1ko cells in later hypoxia reflects a cumulative effect of suppressed early HIF1α stabilisation.

Although we observed a modest decrease (<21%) in HIF1α mRNA levels in GOT1ko cells compared to wild-type cells (Appendix Fig. S6D), treatment of cells with the proteasome inhibitor MG-132 led to similar kinetics of HIF1α protein accumulation in both cell lines (Appendix Fig. S6E), indicating that decreased protein synthesis was unlikely to be the cause for the difference in HIF1α protein levels. We therefore reasoned that the difference in HIF1α stabilisation between wild-type and GOT1ko could be attributable to higher rates of HIF1α degradation in GOT1ko cells. When, after 3 h in hypoxia, HIF1α protein translation was inhibited with cycloheximide, HIF1α protein levels decreased more rapidly in GOT1ko cells than in wild-type MCF7 cells (Fig. 6D). Furthermore, treatment with MG-132 eliminated the difference in HIF1α levels between wild-type and GOT1ko cells but revealed higher levels of hydroxylated HIF1α in GOT1ko than in wild-type MCF7 cells (Fig. 6E; Appendix Fig. S6F). The HIF1α hydroxylation signal was eliminated after treatment with the PHD inhibitor FG-4592, confirming that increased HIF1α hydroxylation is due to PHD activity. Taken together, these data suggest that GOT1ko cells retain higher PHD activity in hypoxia that could account for the delay in HIF1α stabilisation.

Although we observed small differences in mRNA expression of PHDs in GOT1ko compared to wild-type MCF7 cells (Appendix Fig. S6D), such differences were not reflected on the protein level (Appendix Fig. S6G) and pointed to increased PHD activity. PHD activity depends on $O_2$ which, binds to PHDs in an αKG-dependent manner, therefore fluctuations in both $O_2$ and αKG can influence PHD activity. Accordingly, incubation of cells in 1% $O_2$ in the presence of increasing amounts of the cell-permeable αKG analogue dimethyl-αKG (DMKG) led to a dose-dependent decrease in HIF1α protein levels (Appendix Fig. S6H). αKG abundance increased in GOT1ko cells compared to wild-type cells (Fig. 6F),

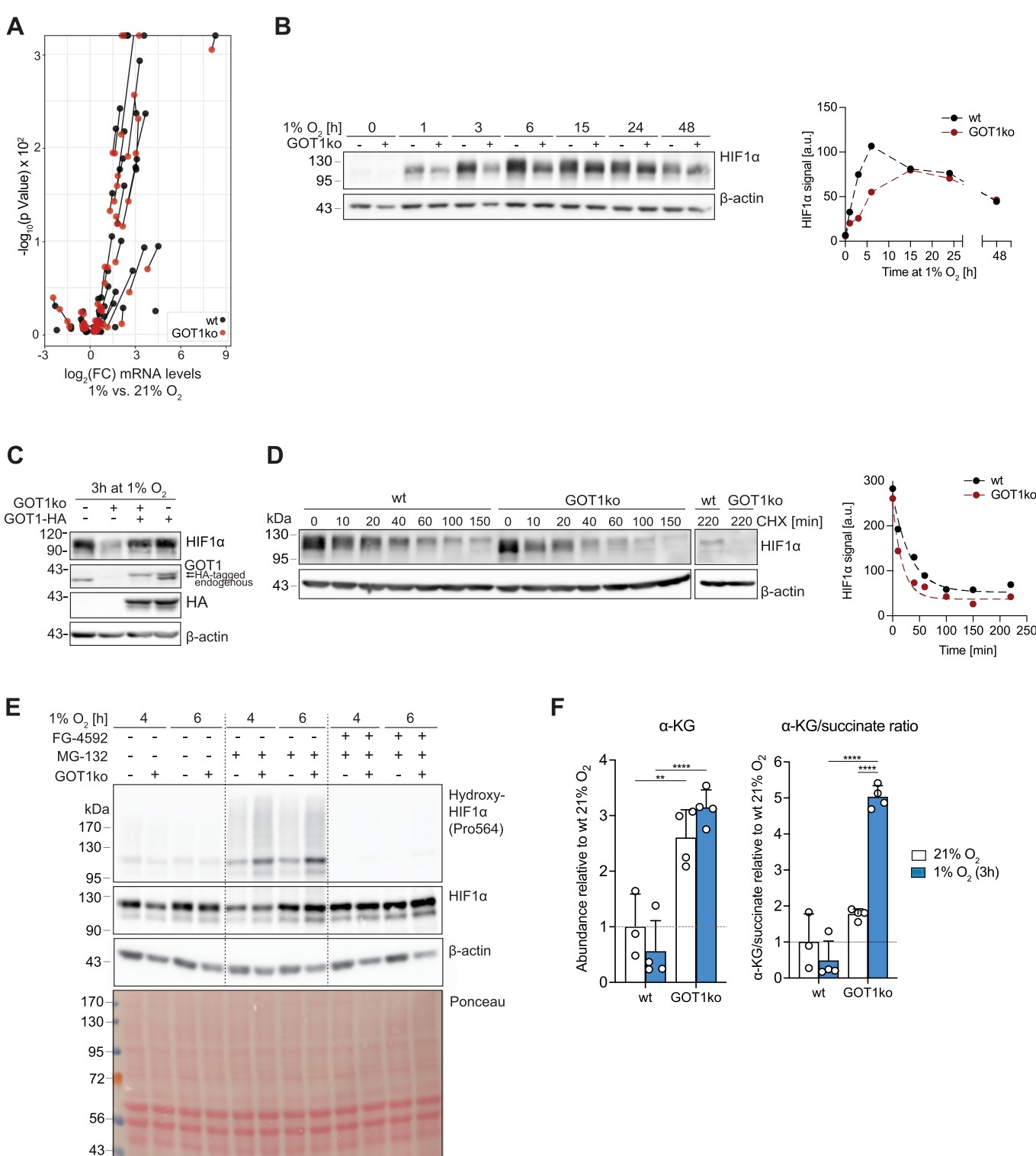

consistent with αKG being a substrate of GOT1, and was associated with an enhanced αKG/succinate ratio, which promotes dioxygenase activity (Hewitson et al, 2007). These data indicate that, in addition to its immediate contribution to glycolysis upon oxygen limitation, GOT1 activity contributes to αKG turnover and thereby controls the kinetics of HIF1α stabilisation.

## Discussion

Adaptation to low oxygen is critical for the survival and proliferation of cancer cells (Bensaad and Harris, 2014). While the PHD-HIF1α signalling axis is a key orchestrator of cellular responses to chronic hypoxia, the mechanisms that allow cells to

**Figure 6. Elevated αKG levels, increased HIF1α hydroxylation and attenuated HIF1α stabilisation in GOT1ko cells under hypoxia.**

(A) Volcano plot of gene expression changes of a panel of HIF1α target genes in wild-type (wt) MCF7 and GOT1ko cells exposed to 1% $O_2$ for 24 h, compared to control cells in normoxia. Lines connect identical genes in the two cell lines to illustrate the differences in hypoxia-induced gene expression changes. (B) Western blot to assess HIF1α protein levels in wild-type (wt) and GOT1ko MCF7 cells exposed to 1% $O_2$ for the indicated lengths of time. The graph on the right shows the quantification of the HIF1α signal. (C) Western blot to assess the protein levels of HIF1α, endogenous GOT1 and HA-tagged GOT1 in wild-type (wt) and GOT1ko MCF7 cells stably expressing GOT1-HA or GFP and exposed to 1% $O_2$ for 3 h. (D) Western blot to assess HIF1α protein levels in wild-type (wt) and GOT1ko MCF7 cells exposed to 1% $O_2$ for 3 h and then treated with cycloheximide (CHX, 20 μM) for the indicated lengths of time. Graph on the right shows the quantification of the HIF1α signal. (E) Western blot to assess the levels of HIF1α, and HIF1α hydroxylated at proline 564 (Pro564) in wild-type (wt) and GOT1ko MCF7 cells exposed to 1% $O_2$ for the indicated lengths of time. Cells were treated with the PHD inhibitor FG-4592 (50 μM), the proteasome inhibitor MG-132 (10 μM) or a combination of both for the duration of the experiment. See also Appendix Fig. S6F for additional controls. (F) The intracellular abundance of α-ketoglutarate (α-KG, left) and corresponding αKG/succinate ratios (right) in wild-type (wt) and GOT1ko MCF7 cells after 3 h at 21% $O_2$ or 1% $O_2$. Data information: Data are representative of experiments with similar conditions performed independently $N$ times as follows: $N = 1$ (A), $N = 3$ (B), $N \geq 3$ (C), $N = 2$ (D–F). Datapoints in (F) represent mean ± s.d. $n = 3$ (A) and $n = 4$ (F) cultures for each cell line and condition. FDRs in (A) were calculated using the 'exactTest' function of the edgeR package (see 'Methods') with a cut-off set at 1%; only changes with FDR < 0.01 are shown. The $P$ values shown in (F) were calculated by two-way ANOVA Sidak's test. ns non-significant, *$P < 0.05$, **$P < 0.01$, ***$P < 0.001$, ****$P < 0.0001$. Source data are available online for this figure.

survive until a full HIF1α response is established are not well understood.

It has long been recognised that oxygen suppresses glycolysis in both healthy and transformed cells, a phenomenon known as the Pasteur effect. Accordingly, we find that, upon oxygen limitation, glycolysis increases within 3 h and demonstrate that this occurs in a HIF1α-independent manner. Various well-established allosteric and signalling mechanisms that increase flux through the first enzymatic steps in glycolysis and glucose transport lead to increased upper glycolysis in hypoxia (Burgman et al, 2001; Clavo et al, 1995; Krebs, 1972; Liemburg-Apers et al, 2016; Morgan et al, 1961; Racker, 1980; Shetty et al, 1993; Stubbs et al, 1972). It is, therefore, reasonable to suggest that such mechanisms elevate the activity of upper glycolysis in early hypoxia and impose an increased requirement for $NAD^+$ that is used by GAPDH to enable the flow of incoming carbons to lower glycolysis.

Lactate labelling from [4-$^2$H]-glucose increased in early hypoxia, in the absence of detectable changes in LDHA protein levels, suggesting that LDHA has reserve capacity in normoxia that can be used in early hypoxia to sustain $NAD^+$. Intriguingly, we also observed increased α-GP synthesis from glucose in early hypoxia, which indicated a shift of the TPI equilibrium towards DHAP, the substrate of the α-GP-producing enzyme GPD1. The TPI equilibrium is influenced by the relative activities of GAPDH (which depends on $NAD^+$ – see scheme in Appendix Figure S4J) and GPD1 (which depends on NADH). In hypoxia, a lower $NAD^+$/NADH ratio, reflecting increased NADH levels, could either inhibit GAPDH (Aithal et al, 1985), promote GPD1 activity (Bentley and Dickinson, 1974) or both. LDHA knock-out led to decreased lower glycolysis and a concomitant increase in α-GP synthesis from glucose. Importantly, the LDHA inhibitor oxamate did not decrease respiration, confirming that the observed changes in α-GP are due to increased production from glucose, rather than decreased consumption due to lower activity of GPD2, an enzyme that converts α-GP to DHAP and provides reduced flavin adenine dinucleotide ($FADH_2$) for mitochondrial respiration. Furthermore, increased α-GP synthesis after treatment of cells with oxamate correlated well with the decrease in ATP levels under the same conditions, which is likely due to attenuated lower glycolysis given that mitochondrial respiration was unchanged. Together, these observations are in line with a model where efflux of glucose carbons from the core glycolytic pathway to α-GP reflects a bottleneck at the GAPDH step.

In early hypoxia, we showed that both mitochondrial respiration (as also previously shown (Chandel et al, 1997)) and aspartate synthesis from glutamine decreased; these are conditions that have been broadly thought to attenuate MAS activity in chronic hypoxia (Birsoy et al, 2015; Eales et al, 2016; Henderson, 1969; Sullivan et al, 2018), in a HIF1α-dependent manner (Melendez-Rodriguez et al, 2019). Given the function of MAS in translocating electrons between cytosolic and mitochondrial NADH, the strong decrease in aspartate production led us to interrogate the role of GOT1 (the canonical aspartate-consuming enzyme of the MAS) in glycolysis during early hypoxia. These investigations revealed that knock-out of GOT1 resulted in decreased flux through MDH1 and caused a lower $NAD^+$/NADH ratio. Furthermore, loss of GOT1 promoted α-GP synthesis from glucose, similar to LDHAko, and attenuated lactate production from glucose only in hypoxia. These data pointed to an increased need for MDH1-derived $NAD^+$ selectively under oxygen-limiting conditions. In view of the decreased aspartate availability, the finding that the aspartate-consuming GOT1 is required for increased glycolysis in early hypoxia was, at first sight, paradoxical. However, we show that the amount of labelled malate produced from [4-$^2$H]-glucose was similar in normoxia and hypoxia. This finding suggests that MDH1 flux is not impaired, even when aspartate levels decrease to less than 30% of those in normoxic cells, unlike the limitation in biomass production due to low aspartate seen in cells in chronic hypoxia or with mitochondrial defects (Altea-Manzano et al, 2022; Birsoy et al, 2015; Garcia-Bermudez et al, 2018). Importantly, overexpression of GOT1 led to increased lactate levels in hypoxia (Fig. 3H). Together, these results support the idea that even when aspartate levels decrease in early hypoxia, they do not become limiting for GOT1-fuelled MDH1 flux, and indicate that MDH1 flux in normoxia (as in other proliferating cells (Wang et al, 2022)) and early hypoxia is, effectively, saturated.

In light of these observations, increased α-GP synthesis from glucose in early hypoxia in wt cells strongly suggests that $NAD^+$ provided by LDHA, even when supplemented by basal GOT1-MDH1 activity and possibly other pathways that support glycolysis in chronic hypoxia (Kim et al, 2019), is not sufficient for the increased amount of glucose carbons from upper glycolysis to flow into lower glycolysis (Fig. 7—early hypoxia). Importantly, a model where increased upper glycolysis due to the Pasteur effect overwhelms GAPDH capacity also elucidates the apparent increase in the reliance of glycolysis on GOT1-MDH1 in hypoxia, even

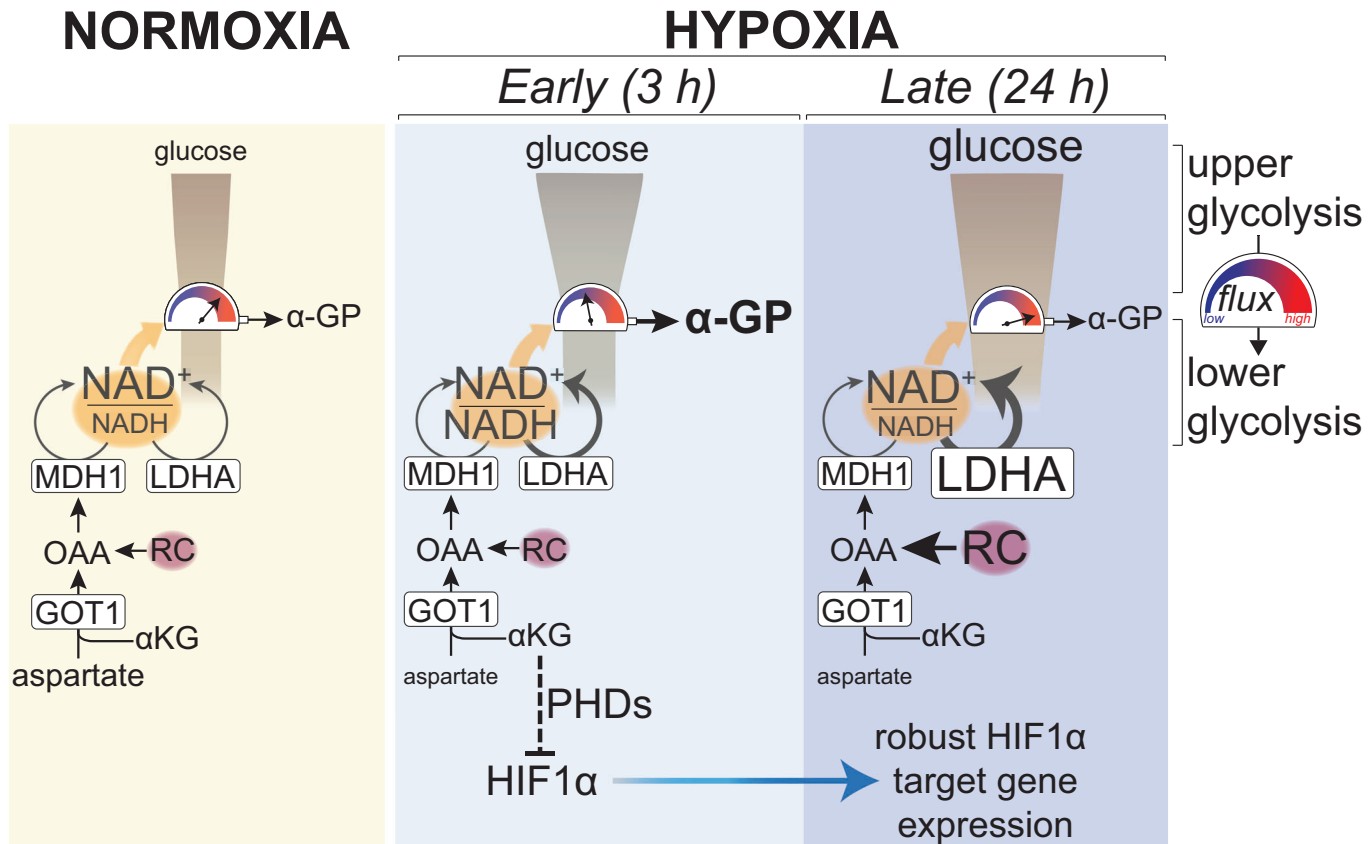

**Figure 7.  Model summarising the dual role of GOT1 in enabling the cellular response to hypoxia.**

In normoxia, carbon flux through lower glycolysis matches that of upper glycolysis because LDHA and GOT1-driven MDH1 provide sufficient NAD$^+$, which is needed for the flow of carbons (indicated by the high reading of the gauge) to lower glycolysis. The coloured scale for the reading of the gauge indicates flux from upper to lower glycolysis. In early hypoxia, elevation of upper glycolysis increases the requirement for regeneration of NAD$^+$, which is supported by an increase in the flux through LDHA and by GOT1-dependent MDH1 activity that does not increase compared to normoxia. However, carbon flow to lower glycolysis is limited by NAD$^+$ in early hypoxia, as indicated by the increased efflux of glucose carbons to α-GP. In late hypoxia, increased RC provides additional OAA for MDH1 and, combined with increased LDHA expression, confers additional NAD$^+$-regenerating capacity enabling increased flow of carbons to lower glycolysis. In parallel, GOT1 consumes αKG (an essential co-factor for PHDs), which, in combination with lower oxygen, suppresses HIF1α hydroxylation and therefore promotes its stabilisation, leading to robust HIF1α target gene expression later in hypoxia. RC reductive carboxylation.

though flux through this pathway is not elevated (Fig. 7). In normoxia, a lower amount of incoming carbons from upper glycolysis can be sustained by sub-saturated LDHA and saturated GOT1-MDH1. However, upon elevation of upper glycolysis in early hypoxia, a greater need for NAD$^+$ arises, requiring "all hands on deck" to provide as much NAD$^+$ as possible, therefore increasing the apparent reliance on GOT1-MDH1. As a consequence, combined inhibition of LDHA and GOT1 is detrimental to cells only in hypoxia but not in normoxia (Fig. 5J), an effect that is associated with an ATP deficit (Fig. 5H). It is likely that, for long-term survival in hypoxia, GOT1-supported cellular bioenergetics work together with other functions of MAS components shown to sustain biomass production in cells subjected to chronic hypoxia or with mitochondrial deficits (Altea-Manzano et al, 2022; Birsoy et al, 2015; Garcia-Bermudez et al, 2018).

In addition to GOT1, another major source of cytoplasmic OAA for MDH1 is ATP citrate lyase (ACL). In chronic hypoxia, RC of glutamine can provide carbons for lipids and OAA via ACL. RC-derived OAA supports MDH1-dependent NAD$^+$ generation, which

is required for glycolysis in cells stimulated with growth factors for 24 h (Hanse et al, 2017) and in cells that harbour stable genetic mutations causing mitochondrial dysfunction (Gaude et al, 2018). In the latter case, increased RC is caused by a decreased NAD$^+$/NADH ratio. We also found a significant decrease in the NAD$^+$/NADH ratio after 3 h under hypoxia, however, RC during this time remained unchanged. Nevertheless, RC increased after 24 h, consistent with previous reports that RC is controlled by HIF1α (Gameiro et al, 2013; Metallo et al, 2011). Furthermore, a modest increase in RC in GOT1ko cells did not suffice to rescue the attenuation of glycolysis upon loss of GOT1.

Interestingly, cells with profound mitochondrial defects take up more aspartate from the media compared to isogenic cells with more functional mitochondria, and exogenous aspartate increases secreted lactate in an MDH1-dependent manner (Gaude et al, 2018). However, whether metabolism of endogenous aspartate via GOT1 itself is sufficient for glycolysis, and the relative contribution of GOT1 and RC to glycolysis remained unexplored. Intriguingly, a higher degree of mitochondrial dysfunction is associated with

decreased flux through the MAS. It has also been shown that reversal of the MAS can overcome the mitochondrial NAD$^+$ deficit in cells with respiratory defects (Altea-Manzano et al, 2022). In such cases, GOT1 provides aspartate to its mitochondrial counterpart, GOT2, which fuels MDH2-dependent NAD$^+$ production to sustain glutaminolysis through glutamate dehydrogenase (GDH). Our data suggest that, in early hypoxia GOT1 largely remains an aspartate consumer and that decreased glutaminolysis is likely due to attenuation of a step upstream of GDH. Altogether, despite parallels between attenuated respiration induced by early hypoxia and respiratory inhibition, the differences highlighted above suggest that RC, and reversal of the MAS, rather than canonical GOT1 activity, may have a more important role for sustaining glycolysis in cells harbouring mitochondrial defects.

Our observation that in early hypoxia glycolytic flux is not maximal, even when supported by both LDHA and GOT1-MDH1, elucidates the need for increased LDHA expression and increased RC to sustain MDH1 in chronic hypoxia. Such a model is further corroborated by our observation that synthesis of α-GP from glucose decreases after 24 h in hypoxia, when both LDHA expression and RC are increased, and are therefore expected to bestow a higher cellular capacity to regenerate glycolytic NADH (Fig. 7—late hypoxia). Combined with evidence from the studies discussed above, our findings highlight the possibility that the mechanisms employed to provide NAD$^+$ to glycolysis, and their relative contributions, may vary depending on different cues (hypoxia, growth factor stimulation, mitochondrial dysfunction) and the different lengths of time they require to increase glycolysis. It is conceivable that combinatorial targeting of multiple redox pathways, as we and others (Hanse et al, 2017) showed, may be a useful therapeutic strategy to attenuate glycolysis. However, the successful selection of relevant target combinations should rely on the relative contributions of these systems to sustaining glycolysis depending on the cellular and physiological context.

Our investigations into the long-term effects of GOT1 knockout also pointed to a role for GOT1 in influencing HIF1α protein levels. While many αKG-consuming and -producing enzymes likely contribute to determining the steady-state intracellular concentration of αKG (Chen et al, 2017; Raffel et al, 2017), our results suggest that GOT1 is a significant consumer of αKG. Given the critical role for αKG in PHD activity, increased αKG in GOT1ko cells is associated with decreased HIF1α protein stability. HIF1α destabilisation may further be promoted by increased cytoplasmic oxygen availability due to decreased respiration (Appendix Fig. S5F), and increased expression of Egln3 (Appendix Fig. S6D) found in GOT1ko cells. Regardless, although HIF1α expression is not completely suppressed, a delay in stabilisation suffices to attenuate HIF1α target gene transcription in the long term, likely due to a cumulative effect over a period of several hours.

Chronic hypoxia develops over periods of tumour growth that are longer than the lengths of hypoxia treatment we used in our experiments. Nevertheless, hypoxia-reoxygenation cycles (also referred to as intermittent hypoxia) have important functional consequences for both tumour physiology and response to therapy (Bhaskara et al, 2012; Brown, 1979; Chaplin et al, 1986; Chen et al, 2018; Gillies and Gatenby, 2007; Jubb et al, 2010; Kang et al, 2020; Toffoli and Michiels, 2008; Verduzco et al, 2015) and occur in timescales that range between minutes and hours (Braun et al,

1999; Brurberg et al, 2006; Cardenas-Navia et al, 2004). Therefore, elucidating what factors determine the ability of cells to rapidly adapt during intermittent hypoxia is crucial. There is significant evidence that other cellular responses to low oxygen, such as suppression of translation (Koritzinsky and Wouters, 2007) and increased ROS production (Hamanaka and Chandel, 2009), can occur within minutes to hours and also play an important role for cell survival. Our findings suggest that GOT1 functions help maintain cells in a primed state that increases their chances of survival when oxygen becomes limiting and, more broadly, support the notion that, upon exposure to stress, cells employ specific mechanisms that help them survive while other adaptive processes, that require more time, are established.

# Methods

## Reagents and tools table

| Reagent/resource | Reference or source | Identifier or catalogue # |
|---|---|---|
| **Antibodies** | | |
| Mouse anti-β-Actin | Sigma-Aldrich | Cat# A2228; RRID:AB_476697 |
| Rabbit anti-GLUT1 | Millipore | Cat# 07-1401; RRID:AB_1587074 |
| Rabbit anti-GAPDH | Cell Signaling Technology | Cat# 2118; RRID:AB_561053 |
| Rabbit anti-GOT1 | Proteintech Group | Cat# 14886-1-AP; RRID:AB_2113630 |
| Rabbit anti-HA, Clone C29F4 | Cell Signaling Technology | Cat# 3724; RRID:AB_1549585 |
| Mouse anti-HIF1α, Clone 54 | BD Biosciences | Cat# 610958; RRID:AB_398271 |
| Rabbit anti-HIF-2α | Abcam | Cat# ab199; RRID:AB_302739 |
| Rabbit anti-HK2, Clone C64G5 | Cell Signaling Technology | Cat# 2867; RRID:AB_2232946 |
| Rabbit anti-hydroxy-HIF1α Pro564, Clone D43B5 | Cell Signaling Technology | Cat# 3434S; RRID:AB_2116958 |
| Rabbit anti-LDHA | Cell Signaling Technology | Cat# 2012S; RRID:AB_2137173 |
| Rabbit anti-MDH1 | Atlas Antibodies | Cat# HPA027296; RRID:AB_10611118 |
| Rabbit anti-PDH2, Clone D31E11 | Cell Signaling Technology | Cat# 4835S; RRID:AB_10561316 |
| Rabbit anti-PDHE-1α pSer293 | Abcam | Cat# ab92696; RRID:AB_10711672 |
| Rabbit anti-PDK1, Clone C47H1 | Cell Signaling Technology | Cat# 3820; RRID:AB_1904078 |
| Rabbit anti-PKM2, Clone D78A4 | Cell Signaling Technology | Cat# 4053; RRID:AB_1904096 |
| Mouse anti-Tubulin, Clone DM1A | Sigma-Aldrich | Cat# T9026; RRID:AB_477593 |
| Goat anti-rabbit IgG antibody conjugated to HRP | Millipore | Cat# AP132P |
| Goat anti-mouse IgG antibody conjugated to HRP | Millipore | Cat# AP127P |
| **Bacterial strains** | | |
| Subcloning efficiency DH5α competent cells | Thermo Fisher Scientific | Cat# 18265017 |

| Reagent/resource | Reference or source | Identifier or catalogue # |
|---|---|---|
| **Chemicals, peptides, and recombinant proteins** | | |
| Acetic acid | Fisher Scientific | Cat# A10360/PB17 |
| Acetonitrile, Optima LC-MS grade | Fisher Scientific | Cat# A955-212 |
| Ammonium bicarbonate | Fisher Scientific | Cat# 10785511 |
| Antimycin A | Sigma-Aldrich | Cat# A8674 |
| BbsI | Thermo Fisher Scientific | Cat# ER1001 |
| β-Mercaptoethanol | Sigma-Aldrich | Cat# M6250 |
| Bromophenol blue | Sigma-Aldrich | Cat# B0126 |
| Bovine serum albumin | Sigma-Aldrich | Cat# A9647 |
| Blasticidin | Millipore | Cat# 203350 |
| N,O-bis(trimetylsilyl)trifluoroacetamide (BSTFA) + 1% trimethylchlorosilane (TMCS) | Sigma-Aldrich | Cat# 33148 |
| Chloroform | Acros Organics | Cat# 390760025 |
| Cholera toxin | Sigma-Aldrich | Cat# C-8052 |
| Crystal violet | Sigma-Aldrich | Cat# C3886 |
| DMEM, high glucose, no glutamine | Thermo Fisher Scientific | Cat# 11960085 |
| DMEM, no glucose, no glutamine, no phenol red | Thermo Fisher Scientific | Cat# A14430 |
| DMEM/F12 | Thermo Fisher Scientific | Cat# 21331046 |
| DMEM-F12 no glutamine, no glucose | Generon | Cat# L0091 |
| Ethanol | Fisher Scientific | Cat# E/650DF/17 |
| EGF | Preprotech | Cat# 100-15 |
| Foetal calf serum | Sigma-Aldrich | Cat# F7524 |
| FG-4592 | Cayman Chemical | Cat# 15294 |
| Fugene HD | Promega | Cat# E2691 |
| Glucose | Sigma-Aldrich | Cat# SLBC6575V |
| Glucose (4-$^2$H) | Omicron Biochemicals | Cat# GLC-035 |
| Glucose ($^{13}$C6) | Sigma-Aldrich | Cat# 389374 |
| Glutamine | Thermo Fisher Scientific | Cat# 25030-081 |
| Glycerol | Sigma-Aldrich | Cat# G5516 |
| Horse serum | Thermo Fisher Scientific | Cat# 16050-122 |
| Glutamine ($^{13}$C5) | Cambridge Isotope Laboratories | Cat# CLM-1822 |

| Reagent/resource | Reference or source | Identifier or catalogue # |
|---|---|---|
| Hydrocortisone | Sigma-Aldrich | Cat# H-0888 |
| Insulin | Sigma-Aldrich | Cat# I-1882 |
| Lactate | Sigma-Aldrich | Cat# L7022 |
| Methanol, Optima LC-MS grade | Fisher Scientific | Cat# A456-212 |
| MG-132 | Sigma-Aldrich | Cat# 474787 |
| MluI | Thermo Fisher Scientific | Cat# FD0564 |
| β-Nicotinamide adenine dinucleotide (NAD) | Sigma-Aldrich | Cat# N1511 |
| β-Nicotinamide adenine dinucleotide, reduced (NADH) | Sigma-Aldrich | Cat# N8129 |
| β-Nicotinamide mononucleotide (NMN) | BioVision | Cat #2733 |
| Nicotinamide riboside (NR) | Cayman Chemical | Cat #23132 |
| Oligomycin | Sigma-Aldrich | Cat# O4876 |
| Penicillin–streptomycin | Thermo Fisher Scientific | Cat# 15140-122 |
| Paraformaldehyde (PFA) | Sigma-Aldrich | Cat# 158127 |
| Polybrene | Sigma-Aldrich | Cat# H9268 |
| Pyromycin dihydrochloride | Sigma-Aldrich | Cat# P7255 |
| Pyridine | Sigma-Aldrich | Cat# 270970 |
| Pyruvate | Sigma-Aldrich | Cat# P5280 |
| Sodium dodecyl sulfate (SDS) | Sigma-Aldrich | Cat# 400036 |
| *Scyllo*-Inositol | Sigma-Aldrich | Cat# I8132 |
| Trizol | Thermo Fisher Scientific | Cat# 15596026 |
| Tween-20 | Sigma-Aldrich | Cat# P7949 |
| Valine ($^{15}$N,$^{13}$C) | Cambridge Isotope Laboratories | Cat# CNLM-442-H-PK |
| WZB-115 | Merck Millipore | Cat# 400036 |
| XhoI | Thermo Fisher Scientific | Cat# FD0694 |
| **Critical commercial assays** | | |
| BCA Protein Assay Kit | Thermo Fisher Scientific | Cat #23225 |
| CellTiter-Glo Luminescent Cell Viability Assay | Promega | Cat# G7570 |
| Glucose Uptake-Glo Assay | Promega | Cat# J1341 |
| **Deposited data** | | |
| RNA sequencing data: wild-type and HIF1α-mutant MCF7 cells in normoxia (21% $O_2$), and in hypoxia (1% $O_2$) for 3 h or 24 h | This study | GEO: GSE122059 |

| Reagent/resource | Reference or source | Identifier or catalogue # |
|---|---|---|
| **Experimental models: cell lines** | | |
| Human: MCF7 | ATCC | Cat# CRL-12584; RRID:CVCL_0031 |
| Human: MCF10A | ATCC | Cat# CRL-10317; RRID:CVCL_0598 |
| Human: HEK-293T | ATCC | Cat# CRL-321; RRID:CVCL_0063 |
| Human: MDA- MC-231 | ATCC | Cat# CRM-HTB-26; RRID:CVCL_0062 |
| Human: BT-474 | ATCC | Cat# HTB-20; RRID:CVCL_0179 |
| **Oligonucleotides** | | |
| Forward primer GOT1 cDNA amplification: cgcacgcgtaccATGGCACCTCCGTCAGTC | This study | N/A |
| Reverse primer GOT1 cDNA amplification: gcgctcgagCTGGATTTTGGTGACTGCTTC | This study | N/A |
| Forward primer LDHA cDNA amplification: cgcacgcgtaccATGGCAACTCTAAAGGATCAG | This study | N/A |
| Reverse primer LDHA cDNA amplification: gcgctcgagAAATTGCAGCTCCTTTTGGATC | This study | N/A |
| Forward primer HIF1α CRISPR sgRNA: caccgTTCTTTACTTCGCCGAGATC | This study | N/A |
| Reverse primer HIF1α CRISPR sgRNA: aaacGATCTCGGCGAAGTAAAGAAc | This study | N/A |
| Forward primer GOT1 CRISPR sgRNA: caccgAGTCTTTGCCGAGGTTCCGC | This study | N/A |
| Reverse primer GOT1 CRISPR sgRNA: aaacGCGGAACCTCGGCAAAGACTc | This study | N/A |
| Forward primer LHDA CRISPR sgRNA: caccgGGCTGGGGCACGTCAGCAAG | This study | N/A |
| Reverse primer LHDA CRISPR sgRNA: aaacCTTGCTGACGTGCCCCAGCC | This study | N/A |
| Forward primer HIF1α knockout validation: TTCCATCTCGTGTTTTTCTTGTTGT | This study | N/A |
| Reverse primer HIF1α knockout validation: CAAAACATTGCGACCACCTTCT | This study | N/A |
| M13 forward primer: TGTAAAACGACGGCCAGT | This study | N/A |
| **Recombinant DNA** | | |
| pMSCV-Peredox-mCherry-NLS | Addgene | Cat# 32385 |
| pSpCas9(BB)-2A-Puro (PX459) V2.0 | Addgene | Cat# 62988 |
| pUC57-LbNOX | Addgene | Cat# 75285 |
| pLenti-HA-IRES-BSD | Origene | Cat# PS100104 |
| pLenti-GFP-P2A-BSD | Origene | Cat# PS100103 |
| pOTB7-GOT1 | Dharmacon | Clone ID: BC000498 |
| pDNR-LIB-LDHA | Dharmacon | Clone ID: BC067223 |
| **Software and algorithms** | | |
| Prism v7.0c | GraphPad Software | N/A |
| Chemstation | Agilent | N/A |
| Masshunter | Agilent | N/A |
| Xcalibur QualBrowser | Thermo Fisher Scientific | N/A |
| Tracefinder v4.1 | Thermo Fisher Scientific | N/A |

## Cell lines and cell culture

Cell lines (MCF7, female, ATCC Cat# CRL-12584, RRID:CVCL_0031; MCF10A, female, ATCC Cat# CRL-10317, RRID:CVCL_0598; HEK-293T, female, ATCC Cat# CRL-3216, RRID:CVCL_0063; MDA-MB-231, female, ATCC Cat# CRM-HTB-26, RRID:CVCL_0062; BT-474, female, ATCC Cat# HTB-20, RRID:CVCL_0179) were obtained from the American Type Culture Collection (ATCC, Manassas, VA, USA). All cell lines were tested mycoplasma-free and cell identity was confirmed by short tandem repeat (STR) profiling by The Francis Crick Institute Cell Services Science Technology Platform. Cells (except MCF10A) were cultured in high-glucose DMEM (Gibco, Cat# 11960085) supplemented with 10% foetal calf serum (FCS), 2 mM L-glutamine and 100 U/mL penicillin/streptomycin in a humidified incubator at 37 °C, 5% $CO_2$. MCF10A cells were cultured in DMEM/F12 (Gibco, Cat# 21331046), supplemented with 5% horse serum, 2 mM L-glutamine, 20 ng/ml EGF (PreproTech), 0.5 µg/ml hydrocortisone, 100 ng/ml cholera toxin, 10 µg/ml insulin and penicillin–streptomycin (Debnath et al, 2003).

Prior to experiments, cells (except MCF10A) were seeded in glucose-free DMEM (Gibco, A1443001), supplemented with 5.5 mM glucose, 10% dialysed FCS (MWCO 3500), 2 mM L-glutamine and penicillin–streptomycin. MCF10A cells were seeded in DMEM/F12 (Generon Cat# L0091) supplemented with 5.5 mM glucose, 5% dialysed horse serum (MWCO 3500), 2 mM L-glutamine, 20 ng/ml EGF (PreproTech), 0.5 µg/ml hydrocortisone, 100 ng/ml cholera toxin, 10 µg/ml insulin and penicillin–streptomycin.

## DNA plasmids and cloning

pMSCV-Peredox-mCherry-NLS (Addgene plasmid # 32385) was a gift from Gary Yellen (Hung et al, 2011); pSpCas9(BB)-2A-Puro (PX459) V2.0 (Addgene plasmid # 62988) was a gift from Feng Zhang (Ran et al, 2013). Plasmids pLenti-HA-IRES-BSD (PS100104) and pLenti-GFP-P2A-BSD (PS100103) were from Origene. pUC57-LbNOX was a gift from Vamsi Mootha (Addgene plasmid # 75285) and used to subclone LbNOX into the MluI and XhoI sites of pLenti-HA-IRES-BSD. The plasmids containing the cDNA of human GOT1 (Clone ID: BC000498) and human LDHA (Clone ID: BC067223) were obtained from Dharmacon. The cDNAs were amplified (GOT1 forward primer: cgc**acgcgt**ac-cATGGCACCTCCGTCAGTC, GOT1 reverse primer: gcg**ctcgag**CTGGATTTTGGTGACTGCTTC; LDHA forward primer: cgc**acgcgt**accATGGCAACTCTAAAGGATCAG; LDHA reverse primer: gcg**ctcgag**AAATTGCAGCTCCTTTTGGATC; lowercase italics=Kozak sequence; lowercase bold=restriction enzyme cleavage site; UPPERCASE=insert) and cloned into pLenti-HA-IRES-BSD using the MluI and XhoI sites for lentivirus production. Results from experiments where this construct was used were compared to a control cell line transduced with virus encoding empty pLenti-GFP-P2A-BSD vector.

## Hypoxia treatments

Hypoxia treatments were performed in a Baker Ruskinn InVivo$_2$ 400 hypoxic workstation at 1% $O_2$, 5% $CO_2$, 37 °C and 70% humidity. Before each experiment, the media of cells were exchanged with media that had been pre-equilibrated at 1% $O_2$ overnight as soon as cells

were transferred to the hypoxia workstation. For hypoxia treatments prior to metabolomics experiments, see 'Stable isotope labelling and metabolite extraction'.

## Virus production and cell transduction

Retroviruses were produced in HEK-293T cells by co-transfecting pMSCV-Peredox-mCherry-NLS and a plasmid containing the amphotropic receptor gene (pHCMV-AmphoEnv) using FuGENE HD Transfection Reagent (Promega). Viral supernatants were harvested 48 h and 72 h after transfection, filtered and supplemented with 4 μg/mL polybrene. Viral supernatants were added to target cells for 6–8 h and cells were allowed to recover for 24 h prior to selection with 1 μg/mL puromycin for at least 5 days.

Lentiviral production and cell transduction were performed as for retroviruses, except that HEK-293T cells were transfected with pLenti-based vectors together with pMD2.G (VSV-G), pMDLg/pRRE (GAG/POL) and pRSV-Rev. Cells were selected with 5 μg/mL blasticidin for at least 5 days.

## Generation of knockout cell lines using CRISPR/Cas9

CRISPR-Cas9 expression constructs were designed and cloned into pSpCas9(BB)-2A-Puro as previously described (Ran et al, 2013) with minor modifications detailed below. CRISPR guide sequences (sgRNAs) were designed using the MIT CRISPR Design Tool (crispr.mit.edu): (HIF1α_forward: caccgTTCTTTACTTCGCCGA-GATC, HIF1α _reverse: aaacGATCTCGGCGAAGTAAAGAAc, GOT1_forward: caccgAGTCTTTGCCGAGGTTCCGC, GOT1_reverse: aaacGCGGAACCTCGGCAAAGACTc; LDHA_forward: caccGGCTGGGGCACGTCAGCAAG, LDHA_reverse: aaacCTTGCTGACGTGCCCCAGCC). Corresponding guide oligonucleotides were mixed, phosphorylated using T4 Polynucleotide Kinase (New England Biolabs) and annealed in a thermocycler using the following programme: 37 °C for 30 min, 95 °C for 5 min, decrease temperature to 25 °C at 0.1 °C/min. The empty Cas9 expression plasmid was linearised using BbsI, ligated to annealed oligonucleotides and transformed into DH5α E. coli. Colonies were tested for successful insertion by colony PCR using the forward primer AATTTCTTGGGTAGTTTGCAGTTTT and the reverse guide oligonucleotide, with an expected band at 150 bp.

MCF7 cells (70–90% confluency) were transfected with the CRISPR-Cas9 expression constructs using FuGENE HD Transfection Reagent (Promega), according to the manufacturer's instructions. The day after transfection 1 μg/mL puromycin was added to the medium for 72 h before cells were seeded at limiting dilutions to obtain monoclonal colonies (500–1000 cells per 15-cm cell culture dish). After 2 weeks, colonies (>100 cells) were isolated and expanded until they could be tested for loss of the target protein by western blot.

## Validation of HIF1α knockout in MCF7 cells

To verify the sequence of mutated HIF1α alleled in HIF1α-mutant MCF7 cells, genomic DNA was extracted using the NucleoSpin Blood kit (Macherey-Nagel), according to the manufacturer's instructions. Exon 2 of the HIF1α gene, which had been targeted using CRISPR/Cas9, was amplified (forward primer: TTCCATCTCG TGTTTTTCTTGTTGT, reverse primer: CAAAACATTGCGA

CCACCTTCT) and PCR products were resolved on a 2% agarose gel. Bands around the expected size (317 bp) were purified and ligated into the pCR4Blunt-TOPO vector using the Zero Blunt TOPO PCR Kit for Sequencing (Thermo Fisher Scientific). Ligation products were transformed into One Shot TOP10 Chemically Competent E. coli cells (Thermo Fisher Scientific). E. coli cells were plates on LB plates containing kanamycin with X-gal (20 μl per plate, 8% w/v in dimethylformamide) and plasmid DNA was amplified and sequenced from 10 blue E. coli colonies (M13 forward primer: TGTAAAACG ACGGCCAGT).

## Cell mass accumulation assay (crystal violet staining)

Cells were seeded in 24-well plates (50,000 cells/well), and after the indicated treatments, cells were washed twice with PBS, fixed with 4% PFA, pH 7.4 (15 min, room temperature), washed with PBS and stained with 0.1% crystal violet in 20% methanol. After staining, cells were washed twice with distilled water (10 min each) and dried. After re-solubilisation in 10% acetic acid, absorbance at 595 nm was measured using a Tecan infinite M1000 Pro plate reader.

## Continuous cell proliferation assay

Cells were seeded the day before the experiment in 96-well plates (black, transparent bottom, Corning, #3606; 9000–12,000 cells/well). After the addition of treatments and/or changing the oxygen concentration in the incubator to 1% $O_2$, cell confluence was monitored using an IncuCyte S3 (Essen Bioscience) by taking phase-contrast images using a 10× objective.

## End-point cell proliferation assay

Cells were seeded the day before the experiment in 96-well plates (black, transparent bottom, Corning, #3606; 9000–12,000 cells/well). 24 h after the addition of treatments and/or changing the oxygen concentration in the incubator to 1% $O_2$, cells were fixed by adding an equal volume (100 μl) of 8% paraformaldehyde, pH 7.4 and incubated for 10 min at room temperature. Cells were washed with PBS and stored in PBS at 4 °C until further processing. Cells were stained with DAPI (4′,6-diamidino-2-phenylindole, 1 μg/ml) for 1 h at room temperature, washed twice with PBS and nuclei counting was performed using an Acumen Explorer eX3 laser scanning microplate cytometer (TTP Labtech).

## Cell lysis and western blotting

Cells were washed twice with PBS and lysed in Laemmli buffer (50 mM Tris-HCl pH 6.8, 1% SDS, 10% glycerol) and stored at −20 °C until further use. Lysed samples were sonicated (2 × 10 s), protein concentration was measured, and samples were boiled at 95 °C for 5 min after the addition of 5% β-mercaptoethanol and bromophenol blue. Samples (20 μg of protein) were resolved by SDS-PAGE and proteins were transferred to nitrocellulose membranes by electroblotting. Membranes were blocked with 5% milk in TBS-T (50 mM Tris-HCl pH 7.5, 150 mM NaCl, 0.05% Tween-20) for 1 h at room temperature and incubated with the primary antibody overnight at 4 °C. Membranes were washed three times with TBS-T and incubated with secondary antibody conjugated to horseradish peroxidase (1:2000) in 5% milk TBS-T for 1 h at room

temperature. Antibodies were visualised by chemiluminescence and imaged using medical X-ray film developed in an AGFA Curix 60 processor (Figs. 2A, 3A and 5A; S2A) or imaged using the Amersham Imagequant 600 RGB (Figs. 3F and 6B–D; S4E and S6D-F). Primary antibodies used: mouse anti-β-actin antibody (Sigma-Aldrich Cat# A2228, RRID:AB_476697), 1:2000 in 5% BSA/TBS-T; rabbit anti-GAPDH (Cell Signaling Technology Cat# 2118, RRID:AB_561053), 1:1000 in 5% BSA /TBS-T; rabbit anti-GLUT1 (Millipore Cat# 07-1401, RRID:AB_1587074), 1:5000 in 5% milk /TBS-T; rabbit anti-GOT1 (Proteintech Group Cat# 14886-1-AP, RRID:AB_2113630), 1:250 in 5% BSA /TBS-T; rabbit anti-HA (Clone C29F4, Cell Signaling Technology Cat# 3724, RRID:AB_1549585), 1:1000 in 5% BSA /TBS-T; mouse anti-HIF1α (Clone 54, BD Biosciences Cat# 610958, RRID:AB_398271), 1:250 in 5% milk/TBS-T; rabbit anti-HIF-2α (Abcam Cat# ab199, RRID:AB_302739), 1:1000 in 5% BSA /TBS-T; rabbit anti-HK2 (Clone C64G5, Cell Signaling Technology Cat# 2867, RRID:AB_2232946), 1:1000 in 5% BSA /TBS-T; rabbit anti-Hydroxy-HIF1α Pro564 (Clone D43B5, Cell Signaling Technology Cat# 3434S, RRID:AB_2116958), 1:1000 in 5% BSA /TBS-T; rabbit anti-LDHA (Cell Signaling Technology Cat# 2012S, RRID:AB_2137173), 1:1000 in 5% BSA /TBS-T; rabbit anti-MDH1 (Atlas Antibodies Cat# HPA027296, RRID:AB_10611118), 1:500 in 5% BSA /TBS-T; rabbit anti-PDH2 (clone D31E11, Cell Signaling Technology Cat# 4835S, RRID:AB_10561316), 1:500 in 5% BSA /TBS-T; rabbit anti-PDHE-1α pSer293 (Abcam Cat# ab92696, RRID:AB_10711672), 1:500 in 5% BSA /TBS-T; rabbit anti-PDK1 (Clone C47H1, Cell Signaling Technology Cat# 3820, RRID:AB_1904078), 1:100 in 5% BSA /TBS-T; rabbit anti-PKM2 (Clone D78A4, Cell Signaling Technology Cat# 4053, RRID:AB_1904096), 1:1000 in 5% BSA /TBS-T; mouse anti-Tubulin (Clone DM1A, Sigma-Aldrich Cat# T9026, RRID:AB_477593), 1:2000 in 5% milk/TBS-T; Secondary antibodies (Millipore): Goat anti-rabbit IgG antibody conjugated to HRP, goat anti-mouse IgG antibody conjugated to HRP.

## Transcriptional profiling by RNA sequencing

Cells were seeded 48 h before harvest ($1.5 \times 10^6$ cells per 6-cm plate) and after indicated treatments (3 h or 24 h at 1% $O_2$, or medium change 3 h before harvest in control cells), cells were washed three times with PBS and lysed in 1 ml TRIzol Reagent (Thermo Fisher Scientific). Chloroform (0.2 ml, Acros Organics) was added to lysates and after shaking for 5 s samples were centrifuged for 18 min at $10,000 \times g$. The upper, aqueous phase was mixed with an equal volume of 100% ethanol (500 μl) and RNA was purified using the RNeasy Kit (Qiagen), according to the manufacturer's instructions. DNase treatment was omitted since initial tests showed no contamination by genomic DNA. After RNA quantification and quality control (NanoDrop, Qubit and Agilent 2100 Bioanalyzer), libraries were prepared using the TruSeq RNA Library Prep Kit v2 (Illumina) or KAPA mRNA HyperPrep Kit (Kapa Biosystems). mRNA sequencing was performed on a Illumina HiSeq 2500 instrument (paired-end or single-end reads, 25 million reads total).

## ATP quantification assay

Cells were seeded in 96-well plates (15,000 cells per) 24–48 h before measurement. After treatment with hypoxia for the indicated lengths of time, or the indicated compounds for 3 h, ATP-dependent luciferase luminescence was measured using the CellTiterGlo kit (Promega, Cat# G7570) according to the manufacturer's instructions. Luminescence counts were normalised to cell number.

## Glucose uptake

Cells were seeded in 96-well plates (15,000 cells per well) 48 h prior to measurement. 2DG uptake was measured using the Glucose Uptake-Glo Assay kit (Promega) after indicated hypoxia treatments (or medium change 3 h before the experiment for control cells) and incubation with 1 mM 2DG for 10 min.

For measurements under hypoxic conditions, 2DG was added to cells in the hypoxia workstation, the plates were sealed with several layers of parafilm, transferred to ambient atmosphere and incubated for 10 min. After adding stop and neutralisation reagents, samples were transferred to a white 96-well plate and incubated for 1 h before luminescence was measured using a Tecan infinite M1000 Pro plate reader. Luminescence counts were normalised to cell number. Wells containing cells without 2DG as well as cells treated with the GLUT1 inhibitor WZB-115 (50 μM for 15 min, #400036, Merck) were used as negative controls.

## Mitochondrial oxygen consumption

Cells were seeded 48 h before the experiment and, after the indicated treatments, cells were trypsinised, centrifuged (1500 rpm, 3 min, room temperature) and resuspended at $5–7.5 \times 10^5$ cells/mL. Hypoxia-treated cells were processed in the hypoxic workstation and resuspended in medium pre-equilibrated in a hypoxic atmosphere (as in Hypoxia treatments). Respiration was measured using an O2k oxygraph (Oroboros instruments) as previously described (Gnaiger, 2008; Gnaiger, 2012) in sealed chambers to prevent reoxygenation of hypoxia-treated cells. Residual oxygen consumption (ROX) was measured after the addition of the complex III inhibitor antimycin A (2.5 μM) and was subtracted from basal oxygen consumption. Oxygen consumption was corrected for cell number.

## Imaging of cytoplasmic NADH with Peredox

MCF7 cells stably expressing Peredox (Hung et al, 2011) were seeded in glass-bottom 24-well plates the day before the experiment (100,000 cells/well). One hour before the experiment, the medium was replaced with imaging buffer (10 mM HEPES pH 7.4, 140 mM NaCl, 1 mM $CaCl_2$, 1 mM $MgCl_2$, 5.4 mM KCl) containing 5.5 mM glucose and 2 mM L-glutamine to acquire baseline images. After image acquisition, cells were incubated with imaging buffer containing 10 mM lactate or 10 mM pyruvate for 5 min before imaging. Cells were kept at 37 °C throughout the experiment and washed with warm imaging buffer between treatments.

Fluorescence images were acquired using an AxioObserver Z1 microscope (Zeiss) and Plan-Apochromat 20×/0.8 M27 objective (picture size $512 \times 512$ pixels, 0.6× zoom, 600.3-μm pinhole. T-sapphire was excited at 800 nm (10% laser power, gain 800) using a Mai Tai DeepSee laser (Spectra-Physics), and emission was recorded at 525 nm. Images were processed using Fiji software by converting pictures from vendor format to 8-bit tiff format, thresholding and identifying nuclei using the 'Analyse particle' function. Mean intensity per particle was used to calculate changes

of T-sapphire fluorescence in individual nuclei from baseline to treatment with lactate or pyruvate.

## Stable isotope labelling and metabolite extraction

Metabolomics sample preparation, GC-MS data processing and analysis were performed as described in (Grimm et al, 2016). In brief, cells were seeded in 6-well plates ($0.35$–$0.5 \times 10^6$ cells per well, 4–5 replicates per condition) 24–48 h before harvest. One hour before the start of the experiment the medium was refreshed and was again changed to medium containing the isotopically labelled nutrient (5.5 mM [U-$^{13}$C]-glucose, 5.5 mM [4-$^2$H]-glucose or 2 mM [U-$^{13}$C]-glutamine) at the start of the experiment. Cells were washed twice with PBS, immediately quenched with liquid nitrogen and kept on dry ice until extraction. Metabolites were extracted by scraping cells in 500 µl methanol, followed by washing the plate with 250 µl methanol and 250 µl water containing the polar internal standard *scyllo*-inositol (1 nmol per sample). Fractions were combined with 250 µl chloroform containing the apolar internal standard [1-$^{13}$C]-lauric acid (C12:0, 40 nmol per sample). Extracts were vortexed, sonicated for $3 \times 8$ min and incubated at 4 °C overnight. Precipitate was removed by centrifugation (10 min, 18,000×*g*, 4 °C) and phases were separated by adding 500 µl water (resulting in 1:3:3 (v/v) chloroform/methanol/water) and centrifugation (5 min, 18,000×*g*, 4 °C). In parallel, cells from three wells per experimental condition were trypsinised and counted using a Nexcelcom Bioscience Cellometer Auto T4 for subsequent normalisation of data to cell number.

The protocol used to extract NAD$^+$ and NADH was adapted from (Lewis et al, 2014). Briefly, cells were scraped in 250 µl acetonitrile:methanol:20 mM ammonium bicarbonate pH 9.0 (2:2:1 v/v) containing 5 µM $^{15}$N$^{13}$C-valine as internal standard, followed by sonication for $3 \times 8$ min and incubation at 4 °C for 1 h. Precipitate was removed by centrifugation, samples were transferred to glass vials, and immediately analysed by LC-MS.

## Gas chromatography-mass spectrometry (GC-MS)

For GC-MS analysis of intracellular metabolites, aqueous phases of cell extracts were transferred to glass vial inserts. For analysis of metabolites in media, 5 µl of media samples were transferred to glass vial inserts and spiked with 1 nmol *scyllo*-inositol. Samples were dried in a centrifugal evaporator and washed twice with 40 µl methanol followed by drying. Samples were methoxymated (20 µl of 20 mg/mL methoxyamine in pyridine, at room temperature overnight) and derivatised with 20 µl of *N,O*-bis(trimetylsilyl) trifluoroacetamide (BSTFA) + 1% trimethylchlorosilane (TMCS) for at least 1 h. GC-MS analysis of metabolites was performed using Agilent 7890B-5977A and 7890A-5975C systems in splitless injection mode (1 µl of sample, injection temperature 270 °C) with a DB-5MS DuraGuard column, helium as carrier gas and electron impact ionisation. Oven temperature was initially 70 °C (2 min), followed by a temperature increase to 295 °C at 12.5 °C/min and subsequently to 320 °C at 25 °C/min (held for 3 min). Chemstation and MassHunter Workstation software (B.06.00 SP01, Agilent Technologies) was used for metabolite identification and quantification by comparison to the retention times, mass spectra, and responses of known amounts of authentic standards. Internal standards were used to correct for sample losses during phase separation and metabolite abundances were normalised to cell number. See Appendix Table S1 for fragment ions used for metabolite quantification by GC-MS.

## Liquid chromatography-mass spectrometry (LC-MS)

Aqueous phases were transferred to glass vial inserts and dried in a centrifugal evaporator. Dried metabolites were resuspended in 100 µl methanol:water (1:1 v/v). The LC-MS method was adapted from (Zhang et al, 2012). LC-MS analysis was performed using a Dionex UltiMate LC system (Thermo Scientific) with a ZIC-pHILIC column (150 mm × 4.6 mm, 5-µm particle, Merck Sequant). A 15 min elution gradient of 80% Solvent A (20 mM ammonium carbonate in Optima HPLC grade water, Sigma-Aldrich) to 20% Solvent B (acetonitrile Optima HPLC grade, Sigma-Aldrich) was used, followed by a 5 min wash of 95:5 Solvent A to Solvent B and 5 min re-equilibration. Other parameters were as follows: flow rate, 300 µL/min; column temperature, 25 °C; injection volume, 10 µL; autosampler temperature, 4 °C.

MS was performed in positive/negative polarity switching mode using a Q Exactive Orbitrap instrument (Thermo Scientific) with a HESI II (Heated electrospray ionisation) probe. MS parameters were as follows: spray voltage 3.5 kV for positive mode and 3.2 kV for negative mode; probe temperature, 320 °C; sheath gas, 30 arbitrary units; auxiliary gas, 5 arbitrary units; full scan range: 70–1050 *m/z* with settings of AGC target and resolution as 'Balanced' and 'High' ($3 \times 10^6$ and 70,000), respectively. Data were recorded using Xcalibur 3.0.63 software (Thermo Scientific). Mass calibration was performed for both ESI polarities before, and lock-mass correction was applied to each analytical run using ubiquitous low-mass contaminants to enhance calibration stability. Pooled biological quality control (PBQC) samples were prepared by pooling equal volumes of each sample, and were analysed throughout the run to provide a measurement of the stability and performance of the instrument. PBQC samples were also analysed in parallel reaction monitoring (PRM) mode to confirm the identification of metabolites. PRM acquisition parameters: resolution 17,500, auto gain control target $2 \times 10^5$, maximum isolation time 100 ms, isolation window *m/z* 0.4; collision energies were set individually in HCD (high-energy collisional dissociation) mode. Qualitative and quantitative analyses were performed using Xcalibur Qual Browser and Tracefinder 4.1 software (Thermo Scientific) according to the manufacturer's workflows. See Appendix Table S2 for fragment ions used for metabolite quantification by LC-MS.

## Quantification and statistical analyses

Statistical analyses were performed using R (R Development Core Team, 2016) or GraphPad Prism v7.0c. Comparisons were made using either two-tailed, unpaired *t* tests, one-way or two-way ANOVA with Dunnett's correction, Sidak's correction or Tukey's correction for multiple comparisons, as indicated in the respective figure legends. All error bars shown in graphs and measurement error in the text (±) represent standard deviation. Statistical errors were propagated where indicated in the figure legends. Significance levels are defined as follows: *$P < 0.05$, **$P < 0.01$, ***$P < 0.001$, ****$P < 0.0001$. Curve-fitting was performed using GraphPad Prism v7.0c using a one-phase decay model with standard parameters. Western blot images were quantified using Fiji ImageJ v1.45 software.

GC-MS metabolomics data were analysed using Agilent Chemstation and MassHunter software, as well as in-house R scripts. Qualitative and quantitative analysis of LC-MS metabolomics data was performed using Xcalibur QualBrowser and Tracefinder 4.1 software (Thermo Fisher Scientific). Isotopic labelling data were corrected for natural isotope abundance using a script provided by Sean O'Callaghan (Bio21 Institute, The University of Melbourne). Where applicable, stripping correction was applied manually. To plot heatmaps, data were expressed as $\log_2$-fold change relative to the mean of the control condition, averaged across all replicates per sample group and subsequently plotted using the RColorBewer v1.1-2 (Neuwirth, 2014) and pheatmap R packages v1.0.8 (Kolde, 2015). Breakpoints for all heatmaps were specified manually to <floor>, $-1.5$, $-1$, $-0.75$, $-0.5$, $-0.25$, 0.25, 0.5, 0.75, 1, 1.5, <ceiling> to enable comparisons across graphs. Z-scores were calculated by subtracting the mean of control condition from the measured value and dividing by the standard deviation of the control condition. All isotope labelling data are expressed as fraction of labelled molecules per metabolite, unless specified otherwise in the figure legends.

Analysis of RNA sequencing data was performed in the R environment and controlled by a GNU make pipeline. Transcript reads were aligned to the Ensembl GRCh37 genome using Tophat2 v2.1.1 (Kim et al, 2013). Aligned transcript reads were filtered for genes with at least 10 reads per gene in 5 or more samples. Between-sample normalisation was performed using the RUVSeq R package v1.10.0 (Risso et al, 2014) and differential expression between sample groups was evaluated using the DESeq2 package v1.16.1 (Love et al, 2014). The 'exactTest' function of the edgeR package v3.18.1 (Robinson et al, 2010) was used to calculate false discovery rates (FDR) and a cut-off of 1% was applied. Ensembl IDs were converted to gene names and Entrez IDs using the AnnotationDBI v1.40.0 (Pagès et al, 2018), EnsDb.Hsapiens.v79 v2.99.0, ensembldb v2.2.2 (Rainer, 2018) and org.Hs.eg.db v3.5.0 (Carlson, 2018) packages.

No blinding, sample size estimation, determination of data normality or randomisation were applied except for metabolomics sample runs, where samples were analysed in a random order, determined using either the RAND function in Microsoft Excel or within the manufacturer's data acquisition software.

## Data availability

The following dataset produced in this study is available: RNA-Seq data: Gene Expression Omnibus GSE122059.

## Peer review information

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

## Acknowledgements

The authors would like to thank all members of the Anastasiou lab for valuable discussions and input throughout this work. The authors thank Mike Howell (Crick High Throughput Screening Science Technology Platform) for advice and help with cell proliferation and viability measurements; Hiroshi Kondo, Erik Sahai (Crick Tumour Cell Biology Laboratory) and Kurt Anderson (Crick Advanced Light Microscopy Science Technology Platform) for help and advice with microscopy-based NADH measurements; the staff at the Crick Advanced Sequencing STP for help with RNA sequencing. The authors acknowledge Sean O'Callaghan (Bio21 Institute, The University of Melbourne) for the algorithm to correct for natural isotope abundance in metabolomics data. The authors also thank Alex Gould and Pia Ballschmieter for critical reading of the manuscript. pMSCV-Peredox-mCherry-NLS was a gift from Gary Yellen (Addgene plasmid # 32385), SpCas9(BB)-2A-Puro (PX459) V2.0 was a gift from Feng Zhang (Addgene plasmid # 62988). pUC57-*Lb*NOX was a gift from Vamsi Mootha (Addgene plasmid # 75285). Work by AJ was supported by the MetaRNA Marie Skłodowska-Curie Innovative Training Network (642738). AA was supported by a Fellowship awarded by The Francis Crick Institute,

GlaxoSmithKline Argentina and the Ministry of Science, Technology and Innovation (MINCYT) of the Republic of Argentina. Funding for this work was provided to DA by the MRC (MC_UP_1202/1) and by the Francis Crick Institute, which receives its core funding from Cancer Research UK (CC2113), the UK Medical Research Council (CC2113) and the Wellcome Trust (CC2113). For the purpose of Open Access, the authors have applied a CC-BY public copyright licence to any Author Accepted Manuscript version arising from this submission.

## Author contributions

**Fiona Grimm**: Data curation; Software; Formal analysis; Investigation; Visualisation; Methodology; Writing—original draft. **Agustín Asuaje**: Data curation; Formal analysis; Validation; Investigation; Methodology. **Aakriti Jain**: Investigation; Methodology. **Mariana Silva dos Santos**: Methodology. **Jens Kleinjung**: Gene expression analysis. **Patrícia M Nunes**: Investigation; Methodology. **Stefanie Gehrig**: Methodology. **Louise Fets**: Investigation; Methodology. **Salihanur Darici**: Investigation. **James I MacRae**: Resources; Methodology. **Dimitrios Anastasiou**: Conceptualisation; Supervision; Funding acquisition; Visualisation; Methodology; Writing—original draft; Project administration; Writing—review and editing.

## Funding

## Disclosure and competing interests statement

The authors declare no competing interests.

