## [Peer Review File · The EMBO Journal]

Metabolic priming by multiple enzyme systems supports glycolysis, HIF1 α stabilisation, and human cancer cell survival in early hypoxia

Fiona Grimm, Agustín Asuaje, Aakriti Jain, Mariana Silva dos Santos, Jens Kleinjung, Patrícia M. Nunes, Stefanie Gehrig, Louise Fets, Salihanur Darici, James I. MacRae, Dimitrios Anastasiou

Corresponding author: Dimitrios Anastasiou (dimitrios.anastasiou@crick.ac.uk)

Review Timeline:	Transfer from Review Commons:	21st Dec 23
	Editorial Decision:	20th Jan 24
	Revision Received:	8th Feb 24
	Accepted:	15th Feb 24

Editor: Kelly Anderson

Transaction Report:

Review
COMMONS

This manuscript was transferred to The EMBO Journal following peer review at Review Commons.

Review #1

1. Evidence, reproducibility and clarity:

Evidence, reproducibility and clarity (Required)

In the paper entitled GOT1 primes the cellular response to hypoxia by supporting glycolysis and HIF1 α stabilization, Grimm and co-authors investigate the metabolic adaptations of cancer cells upon acute hypoxia. By measuring metabolite levels at early time points upon hypoxia, they observe the accumulation of lactate and depletion of aspartate, along with other TCA cycle metabolites. Importantly, they demonstrate that these metabolic changes are independent of the HIF alpha-dependent transcriptional response. The authors investigate the role of aspartate during these initial phases of hypoxia. To this aim, they characterize cells devoid of glutamate oxaloacetate transaminase (GOT1), in which aspartate accumulates and can no longer be used for replenishing NAD⁺ via the downstream conversion of oxaloacetate to malate, via malate dehydrogenase. These cells have lower cytosolic NAD⁺ which affects glycolytic flux through the rate-limiting, NAD⁺-dependent enzyme GAPDH. GOT1 KO cells have a decrease in glucose consumption, lactate secretion and metabolite levels downstream of GAPDH upon early hypoxia, however ATP levels and viability are only affected with additional lactate dehydrogenase (LDH) impairment. Finally, the authors demonstrate that GOT1 KO cells have higher alpha-ketoglutarate (aKG) levels during early hypoxia, which could contribute to higher prolyl-hydroxylation and subsequent degradation of HIF, regulating the transcriptional response mediated by transcription factor.

Major comments

1. The authors claim that they were unable to supplement cells with aspartate (Figure S3), (even though an increase of aspartate is instead observed in cells treated with sodium aspartate) and had to resort to the GOT1 knock-out model to "prevent aspartate from decreasing in hypoxia". This approach implicitly assumes that Got1 is the main driver of aspartate depletion upon hypoxia. However, although steady-state levels of aspartate are indeed higher in these cells, there is still a strong decrease upon hypoxia, which the authors acknowledge but merely ascribe to "attenuated production from glutamine". This seems an insufficient explanation, considering the very fast depletion upon hypoxia originally observed. The authors should provide further information regarding why aspartate is depleted in these conditions and consider other aspartate-consuming enzymes such as GOT2, ASNS, or even nucleotide biosynthesis and urea cycle enzymes. These observations could be made using the labeling experiments already acquired. In addition, to corroborate their hypothesis, the authors could supplement 13 C-aspartate at a supraphysiological concentration (i.e. 5-10 mM) to determine to what extent it is consumed by GOT1 or other pathways.
2. In line with the previous comment, the conclusion that "GOT1 activity, rather than a decrease in aspartate concentration itself, is required to sustain the increase in glycolysis in early hypoxia." seems questionable, especially considering the failed aspartate supplementation. The authors suspect low expression of plasma membrane aspartate transporters as the reason and quote Garcia-Bermudez et al.2018 (PMID: 29941933). This paper contains ranked SLC1A2 mRNA expression data from the Cancer Cell Line

Encyclopedia (CCLE). The authors may apply aspartate supplementation and "early hypoxia" to a cancer cell line expressing SLC1A2 or other aspartate transporters. Alternatively, they could try introducing the transporter by overexpression.

3. The observation that labelled m+1 malate produced from [4-2H]-glucose is similar in normoxia and hypoxia (Figure 4G), does not support the notion that GOT1-MDH axis is increased at low oxygen and seems to suggest that the depletion of aspartate observed in early hypoxia is unrelated to this axis. The authors should resolve this discrepancy.

4. The alpha-KG level regulation by Got1 and the subsequent HIF1alpha "priming" seem quite promising and likely the most novel part of the manuscript. However, further proof should be added to support this strong claim. First, aKG to succinate ratio, rather than aKG alone, is a better indicator of aKG-dependent dioxygenases activity. So, the authors should provide this measurement. Second, the authors should rule out the possibility that the differential hydroxylation of HIF is due to the redistribution of intracellular oxygen due to alterations in mitochondrial function. To do this, they could determine whether cytosolic oxygen levels differ in the two conditions. Finally, the authors could test whether α -ketoglutarate or 2-hydroxyglutarate supplementation affects HIF stability in their experimental conditions.

****Minor comments:****

- The glycerol-3-phosphate shuttle is another means of re-oxidizing NADH and α -GP is indeed higher in GOT1 KO. According to this, in Fig 5C a clear increase in α -GP is observed in LDH KO cells. Would the phenotype be stronger upon additional GPD1 knockout or inhibition?

- Aspartate and lactate levels appear unchanged in MDA-MB231 upon hypoxia. Can these changes be ascribed to a pseudohypoxic state? The authors should comment on this observation.

- Figure S3B: The authors do not provide information on the length of hypoxia for these experiments.

- Glucose and glutamine isotopic labelling should be accompanied by graphs showing the total pool levels of these metabolites, and also the uptake of glucose and glutamine (and their specific isotopologue distribution). It would be important to show the isotopologue distribution of aKG in all the conditions tested, in particular, because of its proposed regulation by Got1.

- Malate generated by MDH1 can be converted by ME1 into Pyruvate, which could be further processed by LDH. Have the authors measured this conversion in their dataset.

- Aspartate absolute levels across cell lines appear different. Is this due to differences in cell volume? Can the authors comment on this observation?

- Under hypoxia the contribution of glutamine (labelled fraction, Fig. S3) to TCA cycle intermediates decreases. However, this is not paralleled by an increase in the contribution of glucose, as also supported by an increase in the m+0 in the glutamine labeling but not in the glucose one. How do the authors explain this apparent inconsistency? Are there sources of unlabelled TCA cycle during the hypoxic experiment?

****Referees cross-commenting****

Referee 2 raises important questions that are in part aligned with referee 1 and are reasonable and doable in the time frame proposed. These are all important questions and comments to

consolidate the central hypothesis of the work and I believe are required for publication.

2. Significance:

Significance (Required)

Overall, this is an exciting and well-executed piece of work focusing on the early biochemical consequences of hypoxia that the wide metabolism/biochemistry audience will appreciate. While most of these observations are not entirely unexpected, the work brings a sufficiently novel perspective and insights to the field and deserves publication. However, some conclusions are not fully supported by the data and some additional experiments are suggested to bring clarification and strengthen the authors' conclusions.

We are a lab expert in cancer metabolism.

3. How much time do you estimate the authors will need to complete the suggested revisions:

Estimated time to Complete Revisions (Required)

(Decision Recommendation)

Between 1 and 3 months

Yes

Review #2

1. Evidence, reproducibility and clarity:

Evidence, reproducibility and clarity (Required)

****Summary****

This manuscript represents an interesting and novel description of the role of a cytosolic transaminase, glutamic-oxaloacetate transaminase 1 (GOT1) on both cytosolic redox (and therefore glycolysis through its functional linkage with malate dehydrogenase 1) and the availability of alpha-ketoglutarate for stabilisation of HIF1a in hypoxia. Some of the most interesting data are the evidence for increased cytosolic NAD⁺ regeneration through the combined action of LDHA (known) and GPD1 (less well-described increase in activity in hypoxia). The manuscript as a whole describes the multiple systems required for the early response to hypoxia, but the focus of the title and way the article is written do not entirely reflect this. For example, the title focuses on GOT1 as the enzymes whose activity is responsible for the early response to hypoxia. However, this is not reflected in some of the data - the deuterium labelling in particular - which shows that LDH and GPD1 are responsible for the biggest redox activity (i.e. support of glycolysis). A degree of reframing of the article may therefore be of benefit.

****Major comments****

1. In Figure 1 C and D, the data suggest significant changes in the decrease in cellular aspartate between 1-2 hours, which then slow. This is followed by a change in lactate concentrations from 2 hours onwards, which is observed in the cells (D) and media (F). The rapid decrease in aspartate concentration suggests a relatively large change, which does not correspond to the later lack of alteration in deuterium labelling from d4-glucose (Figure 4H-J) in m+1 malate. This therefore suggests that the biggest determinant of decreased aspartate is not coupled to MDH1 activity directly. If the manuscript is focused on the relevance of GOT1 activity to the early hypoxic response, this should be better resolved. Given that this could undermine the strength of the case being made for GOT1 activity playing a significant role (through MDH1), could the authors perform the same experiments but in the GOT1KO cells to show how NADH is handled under these conditions by LDHA and GPD1? If the focus of the manuscript is shifted, these experiments would likely not be necessary.
2. The authors present data in Figure 1 and 3 using 2DG as a surrogate for glucose uptake. 2DG has been previously shown not to always be a surrogate for glucose uptake (Sinclair et al. Immunometabolism 2020). Given that this paper highlighted warns in particular about assuming SLC2A1 and SLC2A3 activities based on 2DG uptake, and that these two transporters are the major glucose transporters regulated by hypoxia, a cautious approach to these data is recommended. Assuming that 2DG uptake is a surrogate for glucose in this system (panel C), the effect of GOT1 appears to be at the level of glucose uptake even at 3 hours - it has been marked as being significant by the authors. This suggests that loss of GOT1 has an effect on glucose uptake prior to any transcriptional response is observed. Is the plasma membrane occupancy by the SLC2A1 or SLC2A3 been reduced after GOT1 KO? The same is true for Figure 1 - as intracellular aspartate and lactate and extracellular lactate is shown, could change in extracellular glucose not be presented as a direct measure?
3. The data shown in Figure 2D suggests that there is little change in overall contribution to citrate from glucose in hypoxia compared to normoxia, and that HIF1 does not play a role in the hypoxic response at this point. However, the data presented are overall fractional labelling, and therefore do not focus on the main hypoxia-dependent point of control highlighted before this by the authors - pyruvate oxidation through PDH. Could the authors consider plotting m+2 isotopomer of citrate either alongside or instead of the total fractional label (which includes hypoxia-independent PC activity and cycling carbons). Additionally, the experimental set-up means that average incorporation over the time shown is represented

- i.e. the 3h timepoint is incorporation over the first two hours, while the 24 hour timepoint is averaged over the whole period. It is therefore likely under-representing the decrease in glucose contribution to citrate at 24 hours - the authors could point this out, or OPTIONALLY perform a more time-resolved experiment where flux over shorter periods is assessed for each of the timepoints (i.e. 0-1, 2-3, 5-6, 23-24).

4. Figure 3 data are key for the GOT1 theme of the manuscript, as the authors show that loss of GOT1 increases cellular aspartate in both normoxia and hypoxia - suggesting that GOT1 is an aspartate-consuming enzyme in both conditions. Indeed the magnitude of the change in aspartate after GOT1 knockdown appears similar in both conditions (Panel B). These are interesting data, as they contrast with a recently published study (Altea-Manzano et al. *Molecular Cell* 2022) suggesting that in respiration-deficient cells (a condition with parallels with hypoxia), GOT1 activity may be aspartate producing to supply aspartate to the mitochondria for GOT2. It would be important for the authors to discuss the differences between studies.

5. Panel E shows data at 5 hours, while the rest of the panels here are a mix of 1 and 3h timepoints. Equally panel E also presents concentration, while D presents relative abundance of lactate - could a consistent approach to presenting the results be taken?

6. In Figure S3, the authors show the lack of direct aspartate uptake, or supplementation through the use of an esterified form. OPTIONAL: they could consider using the expression of SLC1A3 (Tajan et al. *Cell Metabolism* 2018; Hart et al *eLife* 2023) to increase aspartate uptake in order to test their hypothesis. Figure S3B-E - the authors suggest based on these data that aspartate decrease in hypoxia is through decreased glutamine contribution. Indeed they could also interrogate the data further, as the defect is observed in glutamate, perhaps suggesting that glutamine metabolism through glutaminase is altered. Figure S3D and E - the authors show data from 3 hours of labelling, which is not at steady-state (observable from the timecourse also shown in B and C). To be able to compare the glucose and glutamine labelling, a timepoint in which (pseudo)steady-state is achieved would be better chose. Additionally, within the aspartate isotopomers arising from glutamine, there is an odd m+1 for aspartate not observed in the other proximal metabolites. Is this a technical defect or is there a biological reason for the significant fractional amount in normoxia?

7. Figure S6F - all samples from GOT1 KO cells have less actin - could an appropriately loaded western blot be presented?

8. In Figure S5B, the authors present ATP data in wild-type control cells, and LDHA-KO with LDHA re-expression. These should be phenotypically similar, but clearly are not. It suggests that there is something not correct with the system being used.

Minor comments

1. PHDs need iron, alpha-ketoglutarate, oxygen and critically ascorbate (Introduction page 2)
2. PDK1 phosphorylation of PDH leads to a reduction in pyruvate oxidation, rather than entry of glucose carbons to the TCA cycle (Introduction page 3)
3. SLC25A51 has been identified as being required for NAD transport into the mitochondria (Kori et al. *Science Advances* 2020), so it is incorrect to say that the inner mitochondrial membrane is impermeable to this metabolite (page 7)
4. Figure S6D - authors shows a highly significant increase in the mRNA for EGLN3, which is a HIF1 target gene, as well as encoding PHD3, which acts to hydroxylate HIF1a alongside

PHD2. This should be commented on in the text.

5. Figure S5G - could it be made clear on the graph whether this is at 21% or 1% O₂?

6. Figure 5I shows ATP level against % labelling of alpha-GP. It isn't clear whether this is abundance or fractional label, but if the latter this is potentially misleading, as if the concentration of alpha-GP increases as fractional label decreases, there is effectively no change. Could the authors extract the steady-state data from the analysis and use this to calculate amount of m+3 label instead of fraction? Similarly for Figure S1H showing fractional labelling of lactate from glucose. It is likely that the title of this graph is a typo, and that m+3 instead was meant. Additionally, measurement of fractional labelling does not demonstrate increased concentrations of the metabolite, but the glucose carbons making up this isotopomer in the pool.

7. Figure S2G - the purpose of the measurement of cysteine is unclear; measurement of NAC directly within cells would be a clearer demonstration of its uptake, and to demonstrate direct contribution to antioxidant response would instead require measurement of cellular antioxidants rather than cysteine itself.

8. There is no Figure S3F (page 6 of text)

9. Figure 2E, lactate excretion into the media is presenting an odd profile, suggesting that between 3 and 6 hour there is uptake by cells. Equally, the 24 hour timepoint is being presented as $p < 0.01$ for 4 replicates with error bars that cross the mean of one of the values. Could the authors possibly check that this is indeed the case?

2. Significance:

Significance (Required)

The data throughout this paper provide some strong evidence for an early and likely HIF-independent metabolic response - while this is understood, detailed studies have not been performed into the various redox balancing cytosolic pathways, which are presented here. The focus on GOT1 is also interesting and novel, but represents part of a larger overall picture presented, which is not reflected in the title.

This is suitable for a relatively broad audience, as the phenotype is likely not cancer specific.

3. How much time do you estimate the authors will need to complete the suggested revisions:

Estimated time to Complete Revisions (Required)

(Decision Recommendation)

Between 3 and 6 months

4. Review Commons values the work of reviewers and encourages them to get credit for their work. Select 'Yes' below to register your reviewing activity at Web of Science Reviewer Recognition Service (formerly Publons); note

that the content of your review will not be visible on Web of Science.

Yes

Review #3

1. Evidence, reproducibility and clarity:

Evidence, reproducibility and clarity (Required)

Here, Grimm and colleagues investigate the immediate cellular response to hypoxia, prior to onset of HIF1 α stabilization/activity. Consistent with established findings they describe that glycolysis is rapidly upregulated under hypoxia, in a HIF1 alpha independent manner, this correlates with an decreased aspartate levels. From this basis, they describe a key role for GOT1 activity in regulating the early hypoxic response, demonstrating its requirement for glycolysis, maintaining the NAD/NADH balance and - in combination with LDHA - maintaining ATP homeostasis in hypoxia. Finally they describe a role for GOT1 (though alpha KG depletion) in contributing to HIF1 alpha stabilization.

In sum, the authors present a compelling study investigating the mechanistic basis of early response to hypoxia, placing GOT1 as a key metabolic regulator of this response. The question of how cell metabolically adapt in the short term to hypoxia is, in my view, an often overlooked area of investigation but clearly has importance across biology, not least in cancer biology - thus the area of investigation is topical. The authors conclusions are supported by their data, often in multiple cell lines and/or through orthologous methods. I would support publication of this study as is.

2. Significance:

Significance (Required)

Significance is stated in my review above, an understudied area of investigation (early hypoxic responses) but clearly important since without a transient response, the long-term impact of HIF1 stress responses would not be possible

3. How much time do you estimate the authors will need to complete the suggested revisions:

Estimated time to Complete Revisions (Required)

(Decision Recommendation)

Cannot tell / Not applicable

No

Full Revision

Manuscript number: RC-2023-02165

Corresponding author(s): Dimitrios Anastasiou

1. General Statements [optional]

We would like to thank the reviewers for their constructive comments that we feel have helped us improve our manuscript. As we outline in our point-by-point responses in detail, and taking into consideration all Reviewers' comments together, in the revised manuscript, we have re-written and expanded parts of the results section to better streamline the justification for investigating GOT1. Also, in response to Reviewer 2's suggestion, we have changed the title and have extensively re-worked the abstract and discussion to better reflect that GOT1 is one of the multiple systems involved in the early response to hypoxia.

Please see overleaf.

Reviewer #1 (Evidence, reproducibility and clarity (Required)):

In the paper entitled GOT1 primes the cellular response to hypoxia by supporting glycolysis and HIF1 α stabilization, Grimm and co-authors investigate the metabolic adaptations of cancer cells upon acute hypoxia. By measuring metabolite levels at early time points upon hypoxia, they observe the accumulation of lactate and depletion of aspartate, along with other TCA cycle metabolites. Importantly, they demonstrate that these metabolic changes are independent of the HIF α -dependent transcriptional response. The authors investigate the role of aspartate during these initial phases of hypoxia. To this aim, they characterize cells devoid of glutamate oxaloacetate transaminase (GOT1), in which aspartate accumulates and can no longer be used for replenishing NAD⁺ via the downstream conversion of oxaloacetate to malate, via malate dehydrogenase. These cells have lower cytosolic NAD⁺ which affects glycolytic flux through the rate-limiting, NAD⁺-dependent enzyme GAPDH. GOT1 KO cells have a decrease in glucose consumption, lactate secretion and metabolite levels downstream of GAPDH upon early hypoxia, however ATP levels and viability are only affected with additional lactate dehydrogenase (LDH) impairment. Finally, the authors demonstrate that GOT1 KO cells have higher alpha-ketoglutarate (aKG) levels during early hypoxia, which could contribute to higher prolyl-hydroxylation and subsequent degradation of HIF, regulating the transcriptional response mediated by transcription factor.

Major comments

1. The authors claim that they were unable to supplement cells with aspartate (Figure S3), (even though an increase of aspartate is instead observed in cells treated with sodium aspartate) and had to resort to the GOT1 knock-out model to "prevent aspartate from decreasing in hypoxia". This approach implicitly assumes that Got1 is the main driver of aspartate depletion upon hypoxia. However, although steady-state levels of aspartate are indeed higher in these cells, there is still a strong decrease upon hypoxia, which the authors acknowledge but merely ascribe to "attenuated production from glutamine". This seems an insufficient explanation, considering the very fast depletion upon hypoxia originally observed. The authors should provide further information regarding why aspartate is depleted in these conditions and consider other aspartate-consuming enzymes such as GOT2, ASNS, or even nucleotide biosynthesis and urea cycle enzymes. These observations could be made using the labeling experiments already acquired. In addition, to corroborate their hypothesis, the authors could supplement ¹³C-aspartate at a supraphysiological concentration (i.e. 5-10 mM) to determine to what extent it is consumed by GOT1 or other pathways.

> We thank the reviewer for this comment that helped us to recognise, in retrospect, that by focusing on GOT1ko as a means to rescue aspartate levels detracted from our main finding and extensive mechanistic insights into the role of GOT1 in sustaining the increase in glycolysis in early hypoxia. As we detail in our response to the Reviewer's point 2, we have now re-written our results section to better clarify why we focused on GOT1 (lines 175-223 of the revised manuscript – please note that line numbering corresponds to the word document with the track changes off). However, we also agree that, because the motivation that led us to GOT1 was the counter-correlation between aspartate and lactate, expanding on the pathways that determine aspartate levels in hypoxia would be useful to the reader.

- To address the reviewer's point, in revised Fig. S3E, we present new data where we incubated cells in normoxia or hypoxia for 3h in the presence of 1.5 mM ¹³C-aspartate. We opted for an intermediate aspartate concentration which was enough to observe intracellular labelling while minimising significant perturbation to cells. We found that the amount of labelled aspartate that accumulates intracellularly is not significantly different between normoxia and hypoxia. At the same time, we observe a vast depletion of unlabelled aspartate. We accept that aspartate labelling may not have reached isotopic steady state within the 3h time point we are confined to for our experiments. However, if increased consumption contributed significantly to aspartate depletion within this timeframe, the amount of labelled aspartate that accumulated would be lower in hypoxia compared to normoxia. Therefore, the data in Fig. S3E indicate that, at least within the timeframe of our experiments, the magnitude of aspartate consumption is not likely to increase to such an extent that could significantly contribute to the depletion in aspartate.

We had, indeed considered other aspartate-consuming pathways, however, in light of the above results and our subsequent finding that GOT1 is needed for increased glycolysis, we did not pursue these investigations any further and focused on the role of GOT1 instead.

- In revised Figure S3, and also in response to one of the Reviewer's other comments below, we have now replotted the data from the experiment in the original manuscript to show both absolute and fractional

isotopologue abundances of TCA intermediates from cells labelled with ^{13}C -glucose or ^{13}C -glutamine. Based on these re-plotted data, we find that the amounts of labelled intermediates from both labels decreases; the apparent decrease from glutamine appears greater than that from glucose, likely because glutamine labels more rapidly a greater fraction of TCA intermediates. Moreover, glutamate fractional labelling from glutamine decreases, but modestly increases from glucose over time in hypoxia compared to normoxia. These data raise the possibility that TCA intermediates are diverted to glutamate synthesis. However, as we point out in the revised text, the fact that only glutamine has reached an isotopic steady state by the end of the time course precludes us from making a more accurate quantitative statement and therefore we have refrained from further elaborating on these observations.

Taking the above observations together, in the revised text we do not dismiss increased consumption as a factor in decreased aspartate levels and rather state that “within the timeframe tested, decreased production is a significant contributor to the low aspartate levels in early hypoxia.” (lines 187-188).

2. In line with the previous comment, the conclusion that "GOT1 activity, rather than a decrease in aspartate concentration itself, is required to sustain the increase in glycolysis in early hypoxia." seems questionable, especially considering the failed aspartate supplementation. The authors suspect low expression of plasma membrane aspartate transporters as the reason and quote Garcia-Bermudez et al. 2018 (PMID: 29941933). This paper contains ranked SLC1A2 mRNA expression data from the Cancer Cell Line Encyclopedia (CCLE). The authors may apply aspartate supplementation and "early hypoxia" to a cancer cell line expressing SLC1A2 or other aspartate transporters. Alternatively, they could try introducing the transporter by overexpression.

> We concede that the way we phrased this statement was not ideal and has rightly led to the reviewer's criticism. In particular, referring to a “decrease in aspartate concentration”, could mislead the reader into thinking that we were referring to the *process* of aspartate consumption, rather than the low aspartate levels themselves, which is what we aimed to explore. In the revised text, we now carefully make this distinction; we show new data (Figure S3G) supporting the idea that low aspartate levels are not necessary for increased lactate; we explain that, given the known role of the malate-aspartate shuttle in coordinating redox balance and potentially affecting glycolytic flux, the fact that aspartate didn't appear to be limiting was surprising and we therefore asked whether GOT1, which depends on aspartate, had a role in the increased glycolysis in early hypoxia. Given that GOT1ko attenuated the increase in glycolysis we subsequently focused on the mechanism underlying this observation. In more detail:

As Reviewer 2 noted in point 1 of their review, the increase in lactate became more apparent after 2 h, when aspartate levels had almost reached their minimum. This successive timing of abundance changes raised the possibility that low aspartate levels precede, and possibly drive, the increased lactate. Therefore, we sought to test whether this was the case by preventing depletion of aspartate in hypoxia with exogenous aspartate. We agree that, to address the comment of Reviewer 1 here, overexpression of an aspartate transporter would have been a good way to overcome poor aspartate uptake by MCF7 cells, however, at the time we initiated this study, SLC1A2 was not known as an aspartate transporter. We, therefore, cultured MCF7 cells for several weeks in media containing 0.5 mM aspartate (which is normally absent in our standard media formulation) because we expected that cells would adapt to take up more aspartate. We, thereby, obtained a derivative cell line that we called MCF7^{Asp}. In new Figure S3G, we show that addition of 0.5 mM aspartate in the media of MCF7^{Asp} cells largely prevented the decrease in intracellular aspartate seen in parental MCF7 cells after 3h in 1% O₂; however, the increase in lactate was similar between MCF7 and MCF7^{Asp} cells. These data are consistent with the idea that the low aspartate levels in hypoxia are not the likely cause for the increase in lactate.

As the Reviewer notes in point 3 below, production of malate m+1 from ^2H -glucose does not decrease below the levels found in normoxia (Fig. 4H), even though aspartate levels are depleted (Fig. 1C). Together with the fact that maintaining aspartate levels to near-normoxic levels does not further boost lactate levels (Figure S3G), these findings speak against the notion that the lack of increased GOT1-MDH1 flux is due to insufficient aspartate and are aligned with the idea that the malate-aspartate shuttle is saturated (PMID: 35973426, 21982705).

3. The observation that labelled m+1 malate produced from [4-2H]-glucose is similar in normoxia and hypoxia (Figure 4G), does not support the notion that GOT1-MDH axis is increased at low oxygen and seems to suggest that the depletion of aspartate observed in early hypoxia is unrelated to this axis. The authors should resolve this discrepancy.

> In our manuscript, we do not claim that the flux through the GOT1-MDH1 axis is increased but, instead, we emphasise the fact that, as the reviewer observed, malate labelling from ^2H -glucose is unchanged (e.g. see text in our original manuscript - lines 519-522 of the revised version: “Importantly, a model where increased upper glycolysis due to the Pasteur effect overwhelms GAPDH capacity also elucidates the

apparent increase in the reliance of glycolysis on GOT1-MDH1 in hypoxia, even though flux through this pathway is not elevated.”). As we also detail in our responses to comments 1 and 2, above, in the revised manuscript, we have re-written the discussion to better explain that the reliance on GOT1 in hypoxia is not driven by increased flux through this pathway (which is likely saturated as outline in our response to point 2, above), but rather from the increased demand imposed by the elevation in incoming glucose carbons due to the Pasteur effect (lines 504-531). This is akin to a situation where increased demand for a product drives its price up if the manufacturer does not boost production to increase supply. We hope that the reviewed discussion makes this clearer and addresses the reviewer’s comment.

4. The alpha-KG level regulation by Got1 and the subsequent HIF1alpha "priming" seem quite promising and likely the most novel part of the manuscript. However, further proof should be added to support this strong claim. First, aKG to succinate ratio, rather than aKG alone, is a better indicator of aKG-dependent dioxygenases activity. So, the authors should provide this measurement.

> In line with the reviewer’s excellent suggestion, in the revised manuscript, we added new panel in Figure 6F (discussed in lines 457-458) that shows α KG levels alongside the corresponding α KG/succinate ratios. These data agree with our original interpretation that cofactor levels in GOT1ko cells favour increased dioxygenase activity.

Second, the authors should rule out the possibility that the differential hydroxylation of HIF is due to the redistribution of intracellular oxygen due to alterations in mitochondrial function. To do this, they could determine whether cytosolic oxygen levels differ in the two conditions.

> The reviewer raises the interesting hypothesis that, given the decreased respiration in hypoxic GOT1ko cells, one could expect increased availability of oxygen that could contribute to the destabilisation of HIF1 α . To the best of our knowledge, measuring absolute cytosolic O₂ concentration, particularly in hypoxia, would require specialised equipment [e.g. phosphorescence lifetime imaging (PMID: 26065366), or phosphorescence quenching oxymetry (PMID: 21912692)]; unfortunately, we do not have access to such equipment. In the revised manuscript, we acknowledge the reviewer’s point with added new text in the discussion (lines 576-577).

Finally, the authors could test whether α -ketoglutarate or 2-hydroxyglutarate supplementation affects HIF stability in their experimental conditions.

> We thank the reviewer for this suggestion. In the revised manuscript (new Figure S6H and lines 453-455) we show that addition of DM- α KG, a cell-permeable form of α KG, to the media of MCF7 cells incubated at 1% O₂, decreases HIF1 α protein levels in a dose-dependent manner and, at the highest dose, to a degree comparable to that of GOT1ko cells.

Minor comments:

- The glycerol-3-phosphate shuttle is another means of re-oxidizing NADH and α -GP is indeed higher in GOT1 KO. According to this, in Fig 5C a clear increase in α -GP is observed in LDH KO cells. Would the phenotype be stronger upon additional GPD1 knockout or inhibition?

> The main phenotype of combined LDHA/GOT1 inhibition is a deficit in ATP and decreased cell survival. While increased flux through GPD1 could, indeed, provide more NAD⁺, this would come at the expense of glucose carbons that would otherwise need to flow into lower glycolysis to produce ATP. Consistent with this idea, our data show that, even if GPD1 or other dehydrogenases reoxidise NADH, as would be the case in both the LDHAko and GOT1ko cells where α -GP is elevated, they are not sufficient to compensate for the decrease in LDH and GOT1 activity. Therefore, we did not pursue this hypothesis further.

- Aspartate and lactate levels appear unchanged in MDA-MB231 upon hypoxia. Can these changes be ascribed to a pseudohypoxic state? The authors should comment on this observation.

> In Figure S2A, we show that MDA-MB-231 cells have increased basal levels of HIF1 α compared to the almost undetectable HIF1 α seen in BT474 (same figure, adjacent panel) or MCF7 cells (Figure 2A). We, therefore, agree with the reviewer’s hypothesis that the attenuated changes in aspartate or lactate levels in MDA-MB-231 cells are likely due to a pseudohypoxic state. As this is speculative, we have refrained from elaborating on this point further in the manuscript.

- Figure S3B: The authors do not provide information on the length of hypoxia for these experiments.

> The data shown in original Figure S3B (new Fig. S3A-B) are a time course. Cells were incubated at 21% or 1% O₂ with the respective isotope label for increasing lengths of time, with the longest time point

shown (6h) being the longest time we incubated cells in hypoxia. If the reviewer meant another panel, the length of hypoxia would be 3h unless otherwise stated.

- *Glucose and glutamine isotopic labelling should be accompanied by graphs showing the total pool levels of these metabolites, and also the uptake of glucose and glutamine (and their specific isotopologue distribution). It would be important to show the isotopologue distribution of α KG in all the conditions tested, in particular, because of its proposed regulation by *Got1*.*

> In the revised manuscript, new Fig. S3 panels A-D, we now show absolute and fractional isotopologue distributions for TCA intermediates for both glucose and glutamine labelling. We have omitted showing α KG in this figure as we could not reliably quantify it in the glutamine-labelling experiment. Also, unfortunately, quantification of glutamine in our GC-MS datasets is not reliable due to conversion to 5-oxoproline.

- *Malate generated by *MDH1* can be converted by *ME1* into Pyruvate, which could be further processed by *LDH*. Have the authors measured this conversion in their dataset.*

> In the figure below we labelled cells with [U-¹³C]-glutamine for 3 h at 21% or 1% O₂ and plotted the fractional labelling for all observable isotopologues in malate, pyruvate and lactate. These data show that there is minimal labelling in pyruvate and lactate (<10% and 3% of labelled molecules, respectively), and that this labelling does not change after 3h in hypoxia (malate shown for comparison).

- *Aspartate absolute levels across cell lines appear different. Is this due to differences in cell volume? Can the authors comment on this observation?*

> To address the reviewer's hypothesis, we focused on MCF7 and MDA-MB-231, the two cell lines with the highest and lowest aspartate levels, respectively. The volume of MCF7 is approx. 19% higher than that of MDA-MB-231 (calculated based on cell size data from PMID: 31015463). Based on this calculation, and bearing in mind that cell volume is a good predictor of biomass content (PMID: 18595067), cell volume differences may contribute to, but cannot fully account for the one order of magnitude difference in aspartate abundance we see between these cell lines (Figures 1C and S1A).

The cell lines we used in this manuscript (MCF7, BT474, MDA-MB-231, MCF10A) represent different breast cancer (or untransformed, in the case of MCF10A) cell types, with different oncogenic mutation content (PMID: 17157791, 22460905) and proliferation rates (PMID: 22628656); all these factors can be related to steady-state cellular metabolite levels (PMID: 31015463). In the figure below, we have plotted aspartate abundance data (from PMID: 31068703) in 928 cell lines of various origins. These data show that aspartate levels can differ as much as 2 orders of magnitude between cancer cell lines and about half an order of magnitude between MCF7 and MDA-MB-231 or BT474 (MCF10A was not present in this dataset); they also show that aspartate levels in the three cell lines rank in the same order as in our manuscript (MCF7>BT474>MDA-MB-231), although, it is unclear if cells in this dataset were also cultured in dialysed serum as in ours, so we cannot confidently compare the absolute aspartate measurements between our studies.

In conclusion, we suspect that cell volume differences together with other factors, such as proliferation rates and metabolic network differences may account for the differences in intracellular aspartate levels.

- Under hypoxia the contribution of glutamine (labelled fraction, Fig. S3) to TCA cycle intermediates decreases. However, this is not paralleled by an increase in the contribution of glucose, as also supported by an increase in the $m+0$ in the glutamine labeling but not in the glucose one. How do the authors explain this apparent inconsistency? Are there sources of unlabelled TCA cycle during the hypoxic experiment?

> While glucose and glutamine are the major carbon sources in many cultured cancer cell lines, incl. MCF7 as indicated by the data in Figure S3A-D, other nutrients (such as amino acids, other than glutamine, and fatty acids) can also provide carbons at various points of the TCA cycle. The fact that fractional labelling of glutamate from glutamine is decreased in hypoxia would suggest that the source of decreased contribution of glutamine into the TCA is unlabelled glutamate. We can exclude uptake of exogenous glutamate, because all our metabolic measurements are performed with cells incubated in media without glutamate and supplemented with dialysed serum. However, we observe a modest increase in the fractional labelling from glucose into glutamate (Figure S3A). As glucose labelling into the TCA cycle is not at steady-state even after 5h, it is hard to assess whether, increased labelling from glucose suffices to explain the dilution of glutamine-derived labelling into glutamate a quantitative conclusion but it points to efflux of intermediates out of the TCA cycle (discussed in lines 181-183 of the revised manuscript).

> We thank the reviewer for their time and thoughtful comments that helped us improve the presentation of our work.

*****Referees cross-commenting*****

Referee 2 raises important questions that are in part aligned with referee 1 and are reasonable and doable is the time frame proposed. These are all important questions and comments to consolidate the central hypothesis of the work and I believe are required for publication.

Reviewer #1 (Significance (Required)):

Overall, this is an exciting and well-executed piece of work focusing on the early biochemical consequences of hypoxia that the wide metabolism/biochemistry audience will appreciate. While most of these observations are not entirely unexpected, the work brings a sufficiently novel perspective and insights to the field and deserves publication. However, some conclusions are not fully supported by the data and some additional experiments are suggested to bring clarification and strengthen the authors' conclusions.

We are a lab expert in cancer metabolism.

Reviewer #2 (Evidence, reproducibility and clarity (Required)):*Summary*

This manuscript represents an interesting and novel description of the role of a cytosolic transaminase, glutamic-oxaloacetate transaminase 1 (GOT1) on both cytosolic redox (and therefore glycolysis through its functional linkage with malate dehydrogenase 1) and the availability of alpha-ketoglutarate for stabilisation of HIF1a in hypoxia. Some of the most interesting data are the evidence for increased cytosolic NAD⁺ regeneration through the combined action of LDHA (known) and GPD1 (less well-described increase in activity in hypoxia). The manuscript as a whole describes the multiple systems required for the early response to hypoxia, but the focus of the title and way the article is written do not entirely reflect this. For example, the title focuses on GOT1 as the enzymes whose activity is responsible for the early response to hypoxia. However, this is not reflected in some of the data - the deuterium labelling in particular - which shows that LDH and GPD1 are responsible for the biggest redox activity (i.e. support of glycolysis). A degree of reframing of the article may therefore be of benefit.

> We thank the reviewer for their constructive suggestions. In the revised manuscript, we have re-written the title and the relevant parts of the results section, and we have significantly re-structured the discussion section to reflect the fact that multiple enzyme systems, one of which is GOT1, converge to support the glycolytic increase and cell survival in early hypoxia. Furthermore, in our point-by-point responses, below, we highlight in detail how we have streamlined the way we present our results.

Major comments.

In Figure 1 C and D, the data suggest significant changes in the decrease in cellular aspartate between 1-2 hours, which then slow. This is followed by a change in lactate concentrations from 2 hours onwards, which is observed in the cells (D) and media (F). The rapid decrease in aspartate concentration suggests a relatively large change, which does not correspond to the later lack of alteration in deuterium labelling from d4-glucose (Figure 4H-J) in m+1 malate. This therefore suggests that the biggest determinant of decreased aspartate is not coupled to MDH1 activity directly. If the manuscript is focused on the relevance of GOT1 activity to the early hypoxic response, this should be better resolved. Given that this could undermine the strength of the case being made for GOT1 activity playing a significant role (through MDH1), could the authors perform the same experiments but in the GOT1KO cells to show how NADH is handled under these conditions by LDHA and GPD1? If the focus of the manuscript is shifted, these experiments would likely not be necessary.

> We thank the reviewer for these comments, which, together with those by Reviewer 1, highlighted that the way we presented our results warranted improvement. First, we would like to clarify that by referring to a “decrease in aspartate concentration”, we may have misled the reader into thinking that we were referring to the *process* of aspartate consumption; rather we wanted to explore whether the low aspartate level itself could be causing the increase in lactate. This is because, as the Reviewer points out, the rate of lactate accumulation picked up after aspartate had almost reached its minimum. Furthermore, by not elaborating on the cause of decreased aspartate and by focusing on GOT1ko as a means to rescue aspartate levels implied a hypothesis whereby GOT1 was the main aspartate consumer, thereby detracting from our main finding and extensive mechanistic insights into the role of GOT1 in sustaining the increase in glycolysis in early hypoxia (regardless the contribution of GOT1 activity in the observed depletion of aspartate).

In the revised text, we have re-written parts of the results section to better clarify these points (e.g. lines 175-223 - please note that line numbering corresponds to the word document with the track changes off). In summary, and as detailed below, we explore the glucose and glutamine data further and present new data with ¹³C-Asp, which, together support the idea that decreased aspartate in early hypoxia is largely attributable to decreased synthesis and, to a lesser extent, if at all, to increased degradation. We then explain that, given the known role of the malate-aspartate shuttle in coordinating redox balance and potentially affecting glycolytic flux, we asked whether GOT1, which depends on aspartate, still had a role in the increased glycolysis vis-à-vis the low aspartate levels in early hypoxia. Given that GOT1ko did attenuate the increase in glycolysis we subsequently focused on the mechanism underlying this observation. We have re-structured the discussion, to highlight that GOT1 is one the multiple systems required for survival in early hypoxia. We also explain that the reliance on GOT1 in hypoxia is not driven by increased flux through the GOT1-MDH1 axis (which is likely saturated), but rather from the increased demand imposed by the elevation in incoming glucose carbons due to the Pasteur effect (lines 504-531). A relatable situation is when increased demand for a product drives its price up if the manufacturer does not boost production to increase supply. We hope that the revised text better clarifies these points.

Below, we detail the new experimental evidence/analyses we referred to above:

- In revised Figure S3A-D, we have now replotted the data from the experiments in the original manuscript to show both absolute and fractional isotopologue abundances of TCA intermediates from cells labelled with ^{13}C -glucose or ^{13}C -glutamine. Based on these re-plotted data, we find that the amounts of labelled intermediates from both labels decreases; the apparent decrease from glutamine appears greater than that from glucose, likely because glutamine labels more rapidly a greater fraction of TCA intermediates. Moreover, glutamate fractional labelling from glutamine decreases, but modestly increases from glucose over time in hypoxia compared to normoxia. These data raise the possibility that TCA intermediates are diverted to glutamate synthesis. However, as we point out in the revised text, the fact that only glutamine has reached an isotopic steady state by 5h precludes us from making a more accurate quantitative statement and therefore we have refrained from further elaborating on these observations.

- In revised Fig. S3E, we present new data where we incubated cells in normoxia or hypoxia for 3h in the presence of 1.5 mM ^{13}C -aspartate. We found that the amount of labelled aspartate that accumulates intracellularly is not significantly different between normoxia and hypoxia. At the same time, we observe a vast depletion of unlabelled aspartate. We accept that aspartate labelling may not have reached isotopic steady state within the 3h time point we are confined to for our experiments. However, if increased consumption contributed significantly to aspartate depletion within this timeframe, the amount of labelled aspartate that accumulated would be lower in hypoxia compared to normoxia. Therefore, the data in Fig. S3E indicate that, at least within the timeframe of our experiments, the magnitude of aspartate consumption is not likely to increase to such an extent that could significantly contribute to the depletion in aspartate.

Together with the data in Fig. S3A-D, these findings suggest that decreased aspartate in early hypoxia is to a great degree driven by decreased production.

2. The authors present data in Figure 1 and 3 using 2DG as a surrogate for glucose uptake. 2DG has been previously shown not to always be a surrogate for glucose uptake (Sinclair et al. Immunometabolism 2020). Given that this paper highlighted warns in particular about assuming SLC2A1 and SLC2A3 activities based on 2DG uptake, and that these two transporters are the major glucose transporters regulated by hypoxia, a cautious approach to these data is recommended. Assuming that 2DG uptake is a surrogate for glucose in this system (panel C), the effect of GOT1 appears to be at the level of glucose uptake even at 3 hours - it has been marked as being significant by the authors. This suggests that loss of GOT1 has an effect on glucose uptake prior to any transcriptional response is observed. Is the plasma membrane occupancy by the SLC2A1 or SLC2A3 been reduced after GOT1 KO? The same is true for Figure 1 - as intracellular aspartate and lactate and extracellular lactate is shown, could change in extracellular glucose not be presented as a direct measure?

> The reviewer raises two points: (a) that using 2DG may not faithfully report transporter-mediated glucose uptake and (b) that, if our observations with 2DG are valid, they could point to the possibility that attenuation of glycolysis in GOT1ko cells may be attributable to effects in glucose uptake. In brief, we cannot use glucose measurements in media as an indicator of glucose uptake rates because we do not observe measurable glucose depletion from media within the relevant timeframe (3h) of our experiments.

(a) Given that we did not have access to a set up for using radionuclides, we explored both 2DG-based and glucose depletion from media as potential means to assess glucose uptake. We found that, over 24h, MCF7 cells deplete glucose faster than cells incubated in normoxia for the same amount of time (figure below, A). The magnitude of this increase is similar to that we report using 2-DG (~3-fold, Fig. 1E and 3C). However, we observed only minimal depletion of glucose in the first 3-5 h of culturing cells with fresh media (figure below, B). This is perhaps not surprising given that studies that look at metabolite exchange rates (incl. glucose) typically sample over a period of one to several days rather than hours (e.g. PMID: 31015463, 22628656). In conclusion, we reasoned that detecting a positive change in signal (intracellular 2DG) would provide a more sensitive means than a decrease in extracellular glucose to enable assessment of glucose use within the early time-points that our manuscript is mainly concerned with.

(b) Indeed, we were initially intrigued by the decrease in glucose uptake by GOT1ko cells as it could explain decreased lactate production. However, the upregulation of upstream glycolytic intermediates in GOT1ko cells in both normoxia and hypoxia (Figure 4A) together with the evidence of increased α -GP production from glucose (Figure 4K-L) suggested that, even if less glucose is taken up by GOT1ko cells, there is still a bottleneck at the GAPDH step that prevents maximal flow of glycolytic intermediates to lower glycolysis. We therefore did not pursue further the cause of decreased glucose uptake by GOT1ko cells at this stage.

3. The data shown in Figure 2D suggests that there is little change in overall contribution to citrate from glucose in hypoxia compared to normoxia, and that HIF1 does not play a role in the hypoxic response at this point. However, the data presented are overall fractional labelling, and therefore do not focus on the main hypoxia-dependent point of control highlighted before this by the authors - pyruvate oxidation through PDH. Could the authors consider plotting m+2 isotopomer of citrate either alongside or instead of the total fractional label (which includes hypoxia-independent PC activity and cycling carbons).

> We agree with the reviewer's suggestion. In the revised manuscript, we added a new panel in Fig. 2D that shows the m+2 citrate isotopologue alongside the original fractional labelling data. This new panel is shown as a bar graph to enable the presentation of individual datapoints and statistical test results.

Additionally, the experimental set-up means that average incorporation over the time shown is represented - i.e. the 3h timepoint is incorporation over the first two hours, while the 24 hour timepoint is averaged over the whole period. It is therefore likely under-representing the decrease in glucose contribution to citrate at 24 hours - the authors could point this out, or OPTIONALLY perform a more time-resolved experiment where flux over shorter periods is assessed for each of the timepoints (i.e. 0-1, 2-3, 5-6, 23-24).

> Indeed, we did consider a more time-resolved labelling experiment as the reviewer suggests, however, we decided against this approach as we were concerned that even if we pre-equilibrated the labelling media in hypoxia, it would be challenging to avoid perturbations associated with handling of the cells during addition of the isotopically labelled compound. The new panel in Fig. 2D that shows absolute citrate m+2 abundances should address this point, however, in the revised text (lines 162-164) we added new text that points out this issue.

4. Figure 3 data are key for the GOT1 theme of the manuscript, as the authors show that loss of GOT1 increases cellular aspartate in both normoxia and hypoxia - suggesting that GOT1 is an aspartate-consuming enzyme in both conditions. Indeed the magnitude of the change in aspartate after GOT1 knockdown appears similar in both conditions (Panel B). These are interesting data, as they contrast with a recently published study (Altea-Manzano et al. Molecular Cell 2022) suggesting that in respiration-deficient cells (a condition with parallels with hypoxia), GOT1 activity may be aspartate producing to supply aspartate to the mitochondria for GOT2. It would be important for the authors to discuss the differences between studies.

> Following the reviewer's suggestion, in the revised manuscript (lines 547-556), we have now expanded our previous discussion on the functions of GOT1 in cells with respiration defects.

5. Panel E shows data at 5 hours, while the rest of the panels here are a mix of 1 and 3h timepoints. Equally panel E also presents concentration, while D presents relative abundance of lactate - could a consistent approach to presenting the results be taken?

> We agree. Taking into consideration that the data in this panel show one time point of the full time-course in Figure S3F, and to streamline the presentation of these data, in the revised manuscript, we have moved the time-course graph to the main figure.

6. In Figure S3, the authors show the lack of direct aspartate uptake, or supplementation through the use of an esterified form. OPTIONAL: they could consider using the expression of SLC1A3 (Tajan et al. Cell Metabolism 2018; Hart et al eLife 2023) to increase aspartate uptake in order to test their hypothesis.

> We agree that, to address this point, overexpression of an aspartate transporter would have been a good way to overcome poor aspartate uptake by MCF7 cells, however, at the time we initiated this study, SLC1A2 was not known as an aspartate transporter. We, therefore, cultured MCF7 cells for several

weeks in media containing 0.5 mM aspartate (which is normally absent in our standard media formulation) because we expected that cells would adapt to take up more aspartate. We, thereby, obtained a derivative cell line that we called MCF7^{ASP}. In new Figure S3G, we show that addition of 0.5 mM aspartate in the media of MCF7^{ASP} cells largely prevented the decrease in intracellular aspartate seen in parental MCF7 cells after 3h in 1% O₂. However, the increase in lactate was similar between MCF7 and MCF7^{ASP} cells. These data are consistent with the idea that the low aspartate levels in hypoxia are not the likely cause for the increase in lactate.

Figure S3B-E - the authors suggest based on these data that aspartate decrease in hypoxia is through decreased glutamine contribution. Indeed they could also interrogate the data further, as the defect is observed in glutamate, perhaps suggesting that glutamine metabolism through glutaminase is altered.

> To address the Reviewer's point, in revised Figure S3, we have now replotted the data from the experiment in the original manuscript to show both absolute and fractional isotopologue abundances of TCA intermediates from cells labelled with ¹³C-glucose or ¹³C-glutamine. We have elaborated on these results in our response to point 1, and we re-iterate our conclusions here for the Reviewer's convenience: Based on these re-plotted data, we find that the amounts of labelled intermediates from both labels decreases; the apparent decrease from glutamine appears greater than that from glucose, likely because glutamine labels more rapidly a greater fraction of TCA intermediates. Moreover, glutamate fractional labelling from glutamine decreases but modestly increases from glucose over time in hypoxia compared to normoxia. These data raise the possibility that TCA intermediates are diverted to glutamate synthesis. However, as we point out in the revised text, the fact that only glutamine has reached an isotopic steady state by 5h precludes us from making a more accurate quantitative statement and therefore we have refrained from further elaborating on these observations.

Figure S3D and E - the authors show data from 3 hours of labelling, which is not at steady-state (observable from the timecourse also shown in B and C). To be able to compare the glucose and glutamine labelling, a timepoint in which (pseudo)steady-state is achieved would be better chose.

> In the revised manuscript, this concern is now addressed by showing both absolute and relative isotopologue abundances for all available time points. We agree that quantitative comparison of labelling must be done at steady-state conditions, however, as we also point out in the revised text (lines 180-181), only glutamine reaches isotopic steady state by 5h whereas glucose hasn't.

Additionally, within the aspartate isotopomers arising from glutamine, there is an odd m+1 for aspartate not observed in the other proximal metabolites. Is this a technical defect or is there a biological reason for the significant fractional amount in normoxia?

> We thank the reviewer for pointing this irregularity, which we should have clearly identified as such during proofreading of the manuscript. Probed by the reviewer's comment, we reviewed the corresponding data tables used to plot these data and found that M+1 had exactly the same values as M+0. We then inspected the original data and confirmed that this resulted from an error during the copying of the data from the R-script output data table to GraphPad Prism for plotting (the line containing the replicates for the m+0 isotopologue was pasted again in the line of the M+1 isotopologues). This issue is now obsolete, as, in the revised manuscript Fig S3 new panels A-D, we have replaced the fractional data with detailed absolute and fractional labelling showing all isotopologues. We apologise for this error.

7. Figure S6F - all samples from GOT1 KO cells have less actin - could an appropriately loaded western blot be presented?

> In the revised manuscript, we added a new panel with the Ponceau (27/02/2018) staining of the same membrane used for immunoblotting. This staining shows equal loading between all lanes. It is unclear why despite equal loading, the actin signal differs between the two lines.

8. In Figure S5B, the authors present ATP data in wild-type control cells, and LDHA-KO with LDHA re-expression. These should be phenotypically similar, but clearly are not. It suggests that there is something not correct with the system being used.

> As shown in the western blot of this figure, expression of exogenous LDH only reaches a fraction of endogenous levels, which likely explains the partial, albeit significant, rescue of the ATP depletion observed in the LDHAKO cells. We have not been able to achieve higher LDH expression in our cell preparations that would enable us to address this point further.

Minor comments

1. PHDs need iron, alpha-ketoglutarate, oxygen and critically ascorbate (Introduction page 2)

> We thank the reviewer for highlighting this critical omission. In the revised manuscript, we have now added this information (line 58).

2. PDK1 phosphorylation of PDH leads to a reduction in pyruvate oxidation, rather than entry of glucose carbons to the TCA cycle (Introduction page 3)

> We agree with the reviewer that our wording was not accurate, and, in the revised text, we have re-written this part (lines 72-74): "...[PDK1] catalyses the inhibitory phosphorylation of pyruvate dehydrogenase (PDH), leading to attenuated pyruvate oxidation and, consequently, decreased contribution of glucose-derived carbons into the tricarboxylic acid (TCA) cycle."

3. SLC25A51 has been identified as being required for NAD transport into the mitochondria (Kori *et al.* Science Advances 2020), so it is incorrect to say that the inner mitochondrial membrane is impermeable to this metabolite (page 7)

> We agree that, in light of the Kori *et al.* study, the phrasing in our text presented an outdated view of pyridine nucleotide compartmentalisation. The data in Kory *et al.* support SLC25A51 as a mitochondrial NAD⁺ transporter, however, it is not clear if NADH is also a substrate. Furthermore, as the authors also point out, SLC25A51 has a relatively low affinity for NAD⁺ and therefore unlikely to interfere with the functions of the malate-aspartate shuttle. Taking all this into consideration, in the revised text (line 249), we acknowledge the existence of a low-affinity mitochondrial NAD⁺ transporter and retained the statement about impermeability specifically for NADH.

4. Figure S6D - authors shows a highly significant increase in the mRNA for EGLN3, which is a HIF1 target gene, as well as encoding PHD3, which acts to hydroxylate HIF1α alongside PHD2. This should be commented on in the text.

> In the revised discussion (lines 577-578), we acknowledge that increased PHD3 (together with increased oxygen availability, related to Reviewer 1's comment), may additionally contribute to HIF1α destabilisation. Please note that we have also added new data (Figure S6H) in response to Reviewer 1, where we show that exogenous αKG causes HIF1α destabilisation in hypoxia, further supporting the notion that boosting intracellular αKG, alone, can destabilise HIF1α.

5. Figure S5G - could it be made clear on the graph whether this is at 21% or 1% O₂?

> We thank the reviewer for pointing out this omission. We now state clearly both in the revised corresponding legend (line 937) and revised figure that these data are at 1% O₂.

6. Figure 5I shows ATP level against % labelling of alpha-GP. It isn't clear whether this is abundance or fractional label, but if the latter this it potentially misleading, as if the concentration of alpha-GP increases as fractional label decreases, there is effectively no change. Could the authors extract the steady-state data from the analysis and use this to calculate amount of m+3 label instead of fraction? Similarly for Figure S1H showing fractional labelling of lactate from glucose. It is likely that the title of this graph is a typo, and that m+3 instead was meant. Additionally, measurement of fractional labelling does not demonstrate increased concentrations of the metabolite, but the glucose carbons making up this isotopomer in the pool.

> For Figure 5I, we confirm that what we show is based on abundance of α-GP m+3 labelling from glucose and, in the revised manuscript (line 895), we amended the legend to clarify this important point.

We concede that the way we had originally written this sentence, suggested that we derived our conclusion that increased lactate in media was due to increased glycolysis based solely on the fractional data in Fig. S1H. In the revised manuscript, we have re-phrased the relevant sentence (lines 136-137) to indicate that our conclusion is based on the fractional data, *together* with the total lactate data that we show in Fig. 1F.

For all our GC-MS experiments we used ions that we detected reliably in all our experiments – in the case of lactate this is m/z 117. This is a 2-carbon fragment as indicated in the original legend; the molecular formula of the derivatised fragment is shown in Table S2. In the revised manuscript (line 671) we clarify that this fragment contains carbons 2 and 3 of lactate (which we concluded from experiments where labelling with 3,4-¹³C-glucose (which labels lactate at C1) led to partial decrease in this isotopologue); therefore changes in 117 m+2 indicate changes in glycolysis rather glycolysis and the PPP.

7. *Figure S2G - the purpose of the measurement of cysteine is unclear; measurement of NAC directly within cells would be a clearer demonstration of its uptake, and to demonstrate direct contribution to antioxidant response would instead require measurement of cellular antioxidants rather than cysteine itself.*

> We agree with the reviewer's comment that, ideally, we would have measured antioxidants, however, unfortunately our GC-MS experiments do not detect glutathione; we, therefore, opted to show cysteine as the best available proof that NAC was added to these cells from the same experiments where we measured aspartate and lactate.

8. *There is no Figure S3F (page 6 of text)*

> In the original version of our manuscript we had awkwardly placed Figure S3F at the top right side of the figure due to space limitations, so, understandably, the reviewer may have missed it. In the revised manuscript, we have now moved this panel to the main Figure 3E, to also address the reviewer's point 5, above (presentation of lactate data).

9. *Figure 2E, lactate excretion into the media is presenting an odd profile, suggesting that between 3 and 6 hour there is uptake by cells. Equally, the 24 hour timepoint is being presented as $p < 0.01$ for 4 replicates with error bars that cross the mean of one of the values. Could the authors possibly check that this is indeed the case?*

> The overlap of the error bars arises from error propagation as we report the values at each time point relative to $t=0h$. The statistical difference we reported was calculated on the original values at 24 h alone, so to avoid this discrepancy we have opted for removing the results of this statistical test altogether.

Reviewer #2 (Significance (Required)):

The data throughout this paper provide some strong evidence for an early and likely HIF-independent metabolic response - while this is understood, detailed studies have not been performed into the various redox balancing cytosolic pathways, which are presented here. The focus on GOT1 is also interesting and novel, but represents part of a larger overall picture presented, which is not reflected in the title.

This is suitable for a relatively broad audience, as the phenotype is likely not cancer specific.

Reviewer #3 (Evidence, reproducibility and clarity (Required)):

Here, Grimm and colleagues investigate the immediate cellular response to hypoxia, prior to onset of HIF1 α stabilization/activity. Consistent with established findings they describe that glycolysis is rapidly upregulated under hypoxia, in a HIF1 α independent manner, this correlates with an decreased aspartate levels. From this basis, they describe a key role for GOT1 activity in regulating the early hypoxic response, demonstrating its requirement for glycolysis, maintaining the NAD/NADH balance and - in combination with LDHA - maintaining ATP homeostasis in hypoxia. Finally they describe a role for GOT1 (though α KG depletion) in contributing to HIF1 α stabilization.

In sum, the authors present a compelling study investigating the mechanistic basis of early response to hypoxia, placing GOT1 as a key metabolic regulator of this response. The question of how cell metabolically adapt in the short term to hypoxia is, in my view, an often overlooked area of investigation but clearly has importance across biology, not least in cancer biology - thus the area of investigation is topical. The authors conclusions are supported by their data, often in multiple cell lines and/or through orthologous methods. I would support publication of this study as is.

Reviewer #3 (Significance (Required)):

Significance is stated in my review above, an understudied area of investigation (early hypoxic responses) but clearly important since without a transient response, the long-term impact of HIF1 stress responses would not be possible

> We thank the reviewer for their time assessing our manuscript and for their positive feedback.

Dear Dr. Anastasiou,

Congratulations on a great revision! Overall, the referees have been positive. However, one referee has one remaining concern that we feel will greatly strengthen your important study.

When you submit your revised version, please also take care of the following editorial items and add this also to your point-by-point response:

1. Please include an author checklist
2. All figure files should be removed from the manuscript text file. Main figures should be uploaded as individual, high res figure files in TIFF, EPS, or PDF format. Supplementary figures should be compiled in an appendix file
3. We only allow 5 keywords, please remove one
4. Please review our author guide and reformat the data availability section, please rename to "Data Availability"
5. Please remove the author contribution section from the main manuscript
6. Please review our policy on conflicts of interest and rename this section to "Disclosure and competing interests statement"
7. Please reformat references to alphabetical order and remove DOIs for all published works
8. We require the publication of source data, particularly for electrophoretic gels and blots and graphs, with the aim of making primary data more accessible and transparent to the reader. It would be great if you could provide me with a PDF file per figure that contains the original, uncropped and unprocessed scans of all or key gels used in the figure or for graphs, an Excel spreadsheet with the original data used to generate the graphs. The PDF files should be labeled with the appropriate figure/panel number, and should have molecular weight marker; further annotation could be useful but is not essential. The PDF files will be published online with the article as supplementary "Source Data" files.
9. We do not allow "data not shown" in our publications. Please remove this from p39.
10. We include a synopsis of the paper (see <http://emboj.embopress.org/>). Please provide me with a general summary statement and 3-5 bullet points that capture the key findings of the paper.
11. We also need a summary figure for the synopsis. The size should be 550 wide by 200-440 high (pixels). You can also use something from the figures if that is easier.
12. Please ensure that the manuscript text is in .docx format
13. Please correct the order of the manuscript sections as outlined online.
14. Please compile the 6 supplementary figures into an appendix. The appendix should be in one PDF file with a table of contents (including page numbers. Correct nomenclature for the figure is "Appendix Figure S1" etc
15. The reagent and tools table should be uploaded as a separate document using the template provided in the guide to authors
16. Supporting tables S2 and S3 should be added to the appendix as "Appendix Table S1" and Appendix Table S2". Table S1 is uploaded separately and should be renamed Dataset EV1", with the legend in the excel file moved to a separate sheet and the title (Dataset EV1) added. CTable S4 could also be added to the reagent and tools table as our template contains a section on oligonucleotides.
17. Please provide the URL for GSE122059 dataset in the data availability statement. Reviewer access code for GSE122059 should be provided as well
18. Please note that a separate "Data Information" section is required in the legends of figures 1c-f.
19. Fig2e, 3h S3a-d, S4e - add statistical parameters
20. Please define the annotated p values in the legend of figure 2d, 3b-e, 3g, 4b-e, 4g-l, 5b, 5d-e, 5g-h, 5j, 6f, supplemental: 1a-d, 1g-h, 2c, 3a, 3f-g, 4a, 4c-d, 4g-l, 4k, 5a-c, 5e-g.

21. Please indicate the statistical test used for data analysis in the legend of figure 6a, S6b
22. Please add N information in the legend of figures 1c-d, 5i, S3g-h
23. Please add error bars in the legend of figures 5i, S3g-h
24. Please define the arrow in the legend of S2a.
25. For figure S6F, please provide the original 4 separate images as part of the source data since this is a composite image.

Thank you for the opportunity to consider your work for publication. I look forward to your revision.

Yours sincerely,

Kelly M Anderson, PhD
Editor, The EMBO Journal
k.anderson@embojournal.org

Referee #1:

The authors did a meticulous job in addressing the referee's critiques and the work has improved. There is only one minor issue remaining.

In Fig. 3 the authors show that GOT1 plays a role in regulating glycolysis in early hypoxia even though "no substantial increase in aspartate consumption" in aspartate is detected (Figure S3E). The authors show that there is no difference in m+4 aspartate levels, but it would be important to show if there is any difference in the amount of aspartate that is consumed by GOT1 in hypoxic cells for completeness. Could the authors include a graph on the labelled oxoglutarate or malate levels derived from [U-13]-aspartate?

Referee #2:

I thank the authors for the comprehensive reworking of the manuscript and figures. This manuscript is now a better reflection of the data and will be of interest to a broad readership.

Rev_Com_number: RC-2023-02165

New_manu_number: EMBOJ-2023-116482

Corr_author: Anastasiou

Title: Metabolic priming by multiple enzyme systems supports glycolysis, HIF1 α stabilisation and cell survival in early hypoxia

Referee #1:

The authors did a meticulous job in addressing the referee's critiques and the work has improved. There is only one minor issue remaining.

In Fig. 3 the authors show that GOT1 plays a role in regulating glycolysis in early hypoxia even though "no substantial increase in aspartate consumption" in aspartate is detected (Figure S3E). The authors show that there is no difference in m+4 aspartate levels, but it would be important to show if there is any difference in the amount of aspartate that is consumed by GOT1 in hypoxic cells for completeness. Could the authors include a graph on the labelled oxoglutarate or malate levels derived from [U-13]-aspartate?

> We did not reliably detect labelling into α KG from [U-¹³C]-aspartate, therefore, in revised Fig. S3, new panel E-right, we added the requested data for malate. These data show a decrease in unlabelled malate (M+0) that was statistically significant, and a trend of increased labelled (M+4) malate (not statistically significant, comparing each isotopologue in 21 vs 1% O₂ by 2-way ANOVA Sidak's test). This observation is aligned with our previous conclusion that GOT1 activity contributes to, but is unlikely to account, alone, for the hypoxia-induced decrease in aspartate in wild-type cells (revised manuscript new text lines 274-277).

We thank the reviewer for their feedback and their useful suggestion for completing the presentation of this dataset.

Rev_Com_number: RC-2023-02165

New_manu_number: EMBOJ-2023-116482

Corr_author: Anastasiou

Title: Metabolic priming by multiple enzyme systems supports glycolysis, HIF1 α stabilisation and cell survival in early hypoxia

Dear Dr. Anastasiou,

Congratulations on an excellent manuscript, I am pleased to inform you that your manuscript has been accepted for publication in The EMBO Journal. Thank you for your comprehensive response to the referee concerns and for providing detailed source data. It has been a pleasure to work with you to get this to the acceptance stage.

I will begin the final checks on your manuscript before submitting to the publisher next week. Once at the publisher, it will take about three weeks for your manuscript to be published online. As a reminder, the entire review process, including referee concerns and your point-by-point response, will be available to readers.

I will be in touch throughout the final editorial process until publication. In the meantime, I hope you find time to celebrate!

Warm wishes,
Kelly

Kelly M Anderson, PhD
Editor, The EMBO Journal
k.anderson@embojournal.org

You may qualify for financial assistance for your publication charges - either via a Springer Nature fully open access agreement or an EMBO initiative. Check your eligibility: <https://www.embojournal.org/page/journal/14602075/authorguide#chargesguide>

>>> Please note that it is The EMBO Journal policy for the transcript of the editorial process (containing referee reports and your response letter) to be published as an online supplement to each paper. If you do NOT want this, you will need to inform the Editorial Office via email immediately. More information is available here: https://www.embojournal.org/transparent-process#Review_Process